# Grazing behavior and winter phytoplankton accumulation

Mara Freilich[1], Alexandre Mignot[2], Glenn Flierl[3], and Raffaele Ferrari[3]

[1]MIT-WHOI Joint Program in Oceanography & Applied Ocean Science and Engineering, Cambridge, MA, USA
[2]Mercator Ocean International, Ramonville-Saint-Agne, France
[3]Department of Earth Atmospheric and Planetary Science, Massachusetts Institute of Technology, Cambridge, MA

**Correspondence:** Mara Freilich (maraf@mit.edu)

**Abstract.** Recent observations have shown that phytoplankton biomass increases in the North Atlantic during winter, even when the mixed layer is deepening and light is limited. Current theories suggest that this is due to a release from grazing pressure. Here we demonstrate that the often-used grazing models that are linear at low phytoplankton concentration do not allow for a wintertime increase in phytoplankton biomass. However, certain mathematical formulations of grazing as a function of phytoplankton concentration that are quadratic at low concentrations (or more generally decrease faster than linearly as phytoplankton concentration decreases) can reproduce the fall to spring transition in phytoplankton, including wintertime biomass accumulation. We illustrate this point with a minimal model for the annual cycle of North Atlantic phytoplankton designed to simulate phytoplankton concentration as observed by BioGeoChemical-Argo (BGC-Argo) floats in the North Atlantic. This analysis provides a mathematical framework for assessing hypotheses of phytoplankton bloom formation.

## 1  Introduction

One of the most prominent biological events in the surface ocean is the North Atlantic spring bloom (*Boss et al.*, 2008; *Siegel et al.*, 2014; *Cole et al.*, 2015). Each spring, in an event that is distinctive in satellite ocean color observations (*Siegel et al.*, 2014), there is a rapid accumulation of phytoplankton in the ocean surface layer across the North Atlantic. A bloom occurs when the phytoplankton growth rates are sufficiently faster than the loss rates over a sustained time period (*Sverdrup*, 1953). The large annual cycle in the phytoplankton population in the North Atlantic occurs in the context of large seasonal cycles in atmospheric conditions that drive changes in mixed layer depth, surface irradiance, and upper layer temperature. How these environmental factors interact with ecological processes to produce a bloom is still being debated (*Fischer et al.*, 2014).

The traditional theory of phytoplankton population dynamics in the North Atlantic attributes the spring bloom to the release of phytoplankton from light limitation, which causes phytoplankton growth rates to increase. This has become known as the "critical depth hypothesis" (*Sverdrup*, 1953) because the theory states that phytoplankton can begin to grow when the mixed layer has shoaled sufficiently so that the light-dependent phytoplankton growth terms are larger than the phytoplankton loss terms, which are assumed to be constant throughout the winter and into the spring. This theory is based on the idea that biological and physical processes are inherently coupled. The relative timescales of mixed layer turbulence and biological growth influence the rate of phytoplankton accumulation. Phytoplankton can also be released from light limitation while the

mixed layer is deep if turbulence is temporarily reduced (*Huisman et al.*, 1999; *Taylor and Ferrari*, 2011; *Paparella and Vichi*, 2020).

An alternative hypothesis proposed by *Behrenfeld* (2010) focuses on changes in both loss rates and growth rates. The "disturbance-recovery hypothesis" states that even though phytoplankton growth rates are very low in the wintertime, due primarily to light limitation, loss rates decrease even faster as the mixed layer deepens due to decreasing phytoplankton-

zooplankton encounter rates. This hypothesis was formulated as an explanation of recent observations of increasing phytoplankton stocks in the wintertime (*Behrenfeld*, 2010; *Boss and Behrenfeld*, 2010). Wintertime biomass accumulation is inconsistent with the critical depth hypothesis, which assumes that the winter growth rates are smaller than the constant loss rates.

The critical depth hypothesis and the disturbance-recovery hypothesis differ in their predictions of the evolution of winter

loss rates. Process-level understanding and quantification of phytoplankton population loss rates is challenging, because it is very difficult to directly measure the factors that contribute to loss for the whole population. Phytoplankton are thought to be tightly controlled by grazing and loss processes (*Landry and Calbet*, 2004; *Calbet and Landry*, 2004; *Strom et al.*, 2007; *Evans and Brussaard*, 2012; *Prowe et al.*, 2012). Any accumulation depends on the imbalance betweeen growth and loss processes (*Behrenfeld and Boss*, 2018). Loss due to grazing depends on both the concentration of phytoplankton and zooplankton pop-

ulations and on the many factors that mediate the interactions between phytoplankton and zooplankton such as temperature, light, and species composition (*Chen et al.*, 2012; *Moeller et al.*, 2019; *Strom and Welschmeyer*, 1991). Autonomous measurements from satellites and BGC-Argo floats have made quantification of phytoplankton biomass possible over large spatial and temporal scales (*Siegel et al.*, 2002; *Boss et al.*, 2008; *Mignot et al.*, 2018; *Randelhoff et al.*, 2020; *Hague and Vichi*, 2021). No such equivalent measurements exist for zooplankton populations.

The interactions between phytoplankton and zooplankton can be modeled through mathematical relationships that express the rate of phytoplankton consumption by zooplankton as a function of phytoplankton concentration (*Evans and Parslow*, 1985; *Franks*, 2002). There are many functional responses that are supported by experiments and theory and that have been used to represent grazing in numerical simulations and to interpret observations (*Gentleman et al.*, 2003; *Laufkötter et al.*, 2015). The most commonly used functional responses increase linearly or quadratically and saturate to a constant rate at high

concentrations (*Gentleman et al.*, 2003).

During the spring bloom, phytoplankton accumulation is exponential due to the rapid increase in growth rates that makes loss processes relatively much smaller. In the wintertime, the observed phytoplankton accumulation is slower and leading hypotheses of phytoplankton bloom formation differ in their predictions both of phytoplankton population dynamics and of phytoplankton loss rates. Comparing phytoplankton-zooplankton models with different representations of grazing against the

observations of biomass accumulation during sub-optimal growth conditions, such as during the wintertime, may constrain the range of appropriate grazing functions for winter conditions or even the winter-spring transition. Here, we demonstrate that the disturbance-recovery hypothesis requires a grazing function that decreases more rapidly than linearly at low prey concentrations. We show that a model with a quadratic grazing function at low winter phytoplankton concentrations captures the full annual cycle of phytoplankton biomass in the North Atlantic, i.e. both weak wintertime biomass accumulation and an

explosive springtime bloom. Our aim is to provide empirically motivated guidance for the formulation and testing of grazing models.

## 2 Predator-prey decoupling

In this section we formulate a simple ecosystem model and examine different grazing functions to clarify the relationship between grazing rates and mixed layer depth during winter conditions (Figure 1). Marine planktonic ecosystem dynamics can be

coarsely represented as an interaction between three compartments: nutrients, phytoplankton, and zooplankton. These broad compartments integrate across all the chemical and biological diversity observed in the ocean and are defined by their interactions with each other. In the simple formulation adopted here, the nutrients are consumed by phytoplankton, the zooplankton consume phytoplankton, and the plankton are converted back to nutrients when they die. The set of equations that describe these interactions for the concentrations of nutrients ($n$), phytoplankton ($p$), and zooplankton ($z$) as a function of the ocean

depth $\zeta$ take the form:

$$
\begin{aligned}
\frac{Dn}{Dt} &= -\mu(n,t)e^{K_d\zeta}p + d_p p + (1-a)g(p)z + d_z z^2 + \frac{\partial}{\partial\zeta}\kappa\frac{\partial n}{\partial\zeta}, \\
\frac{Dp}{Dt} &= \mu(n,t)e^{K_d\zeta}p - g(p)z - d_p p + \frac{\partial}{\partial\zeta}\kappa\frac{\partial p}{\partial\zeta}, \\
\frac{Dz}{Dt} &= ag(p)z - d_z z^2 + \frac{\partial}{\partial\zeta}\kappa\frac{\partial z}{\partial\zeta}.
\end{aligned}
\tag{1}
$$

The vertical coordinate, $\zeta$, is zero at ocean surface and negative below. All compartments are mixed in the vertical by ocean turbulence at a rate set by the diffusivity $\kappa$. The phytoplankton specific growth rate depends on nutrients, according to the function $\mu(n)$, and decays exponentially with depth due to the absorption of light with depth with an attenuation coefficient

$K_d$. We model growth as a linear function of light, which reduces the number of parameters required. This choice increases the sensitivity of growth to light at high irradiance relative to a saturating model, but at the low irradiance conditions typical of the wintertime, the focus of this manuscript, growth depends approximately linearly on light (*Franks*, 2002). Phytoplankton mortality (from causes other than grazing by zooplankton), $-d_p p$, is linear in $p$. Zooplankton mortality, $-d_z z^2$, is quadratic in $z$ to account, implicitly, for intratrophic and higher trophic level predation; this choice has the additional property of preventing

extinction of zooplankton in winter. The grazing of phytoplankton by zooplankton is linear in $z$ and proportional to $p$ according to the grazing function $g(p)$. Zooplankton are messy eaters and ingest only a fraction $a < 1$ of $g(p)z$. The grazing function represents a density-dependent mortality process. Other mortality processes such as viral lysis are also believed to be density-dependent and could be studied within the same framework (*Weitz et al.*, 2015; *Mateus*, 2017). While we retrict the analysis to zooplankton grazing, our qualitative conclusions are likely to apply to other density-dependent mortality processes.

To illustrate the importance of the form the grazing term, we will examine the model in equation 1 for the wintertime period through the bloom onset. During this period, we can make a few simplifying assumptions. First, we will assume that turbulence is strong enough to keep all compartments well mixed in the vertical over a mixed layer of depth $H$. This assumption holds if the turbulence mixes all compartments throughout $H$ on a timescale faster than any biological timescale (*Taylor and Ferrari*, 2011). Equivalently, all references to the mixed layer should be interpreted as the actively mixing layer. Second, we will assume

that winter growth is not nutrient limited ($n \gg n_0$ so that $\frac{n}{n+n_0} \to 1$) and thus saturates to a constant mixed layer-averaged

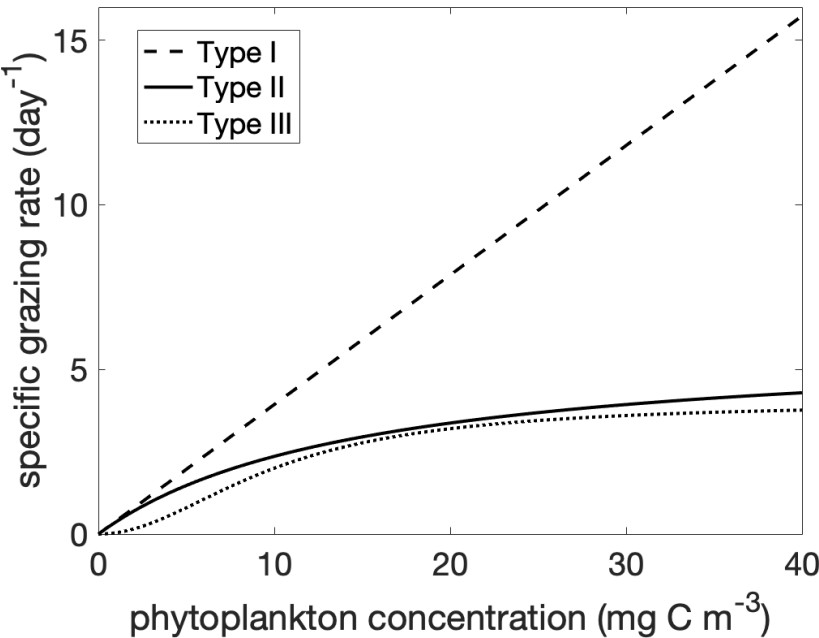

**Figure 1.** Grazing rate $g(p)$ as a function of phytoplankton concentration for Holling Type I, II, and III functional responses. The parameters $g_0$ and $p_0$ are given in Table 1 for the Holling type II and III functional responses. The forms of the Holling type II and III is as in equation 6. The form of the Holling type I is the linearized type II, $g_{H_I}(p) = g_0/p_0 p$.

value $\mu_0$ (we will not make this assumption about a time and nutrient independent growth rate in section 3). Finally, we assume that the mixed layer is deep relative to the depth of light penetration ($HK_d \gg 1$) such that $e^{K_dH}$ is small. All these assumptions are appropriate in winter, the focus of our study, but they are less defensible in other seasons when turbulence is weak (*Taylor and Ferrari*, 2011) and are not all used in section 3.

We formulate a bulk mixed layer model by employing these simplifying assumptions and taking the vertical average of the equations in (1) over the mixed layer depth $H(t)$,

$$\frac{Dp}{Dt} = \left( \frac{1}{K_dH(t)} \left( 1 - e^{K_dH(t)} \right) \mu_0 \frac{n}{n+n_0} - d_p \right) p - g(p)z - s^+p \approx \left( \frac{1}{K_dH(t)}\mu_0 - d_p \right) p - g(p)z - s^+p$$
$$\frac{Dz}{Dt} = ag(p)z - d_zz^2 - s^+z,$$
(2)

where $p$ and $z$ are the constant mixed layer concentrations of phytoplankton and zooplankton, respectively. The term $\mu_0/(K_dH(t))$ is the average growth rate over the mixed layer, which is computed as the integral of the light-dependent growth over the mixed

layer depth divided by the mixed layer depth.

The term $s^+$ appears when taking the vertical average of the mixing term in equation (1). It represents the dilution of phytoplankton and zooplankton that results from the turbulent entrainment of water without biomass across the mixed layer

base and is given by,

$$s^+ = \begin{cases} \frac{1}{H}\frac{dH}{dt} & \frac{dH}{dt} > 0 \\ 0 & \frac{dH}{dt} \leq 0 \end{cases} \tag{3}$$

We can derive an equation for the standing stock of biomass in the mixed layer by taking a vertical integral of the equations in (1). Introducing $P = Hp$ and $Z = Hz$ to represent the total biomass of phytoplankton and zooplankton respectively we have

$$\frac{DP}{Dt} = \left(\frac{1}{K_d H}\mu_0 - d_p\right)P - g(P/H)Z - s^- P$$
$$\frac{DZ}{Dt} = ag(P/H)Z - \frac{1}{H}d_z Z^2 - s^- Z \tag{4}$$

In contrast to the average concentration, the total biomass does not change due to the physical effects of dilution. However, 110 when the mixed layer shoals, biomass is lost below the mixed layer through detrainment and the total biomass decreases at a rate given by $s^-$

$$s^- = \begin{cases} 0 & \frac{dH}{dt} > 0 \\ \frac{1}{H}\frac{dH}{dt} & \frac{dH}{dt} \leq 0 \end{cases} \tag{5}$$

In the following subsections we will analyze the phenology of phytoplankton for different choices of grazing functions (Figure 1). The linear (Holling type I) grazing function assumes that the plankton-specific grazing rate (units of per day) increases 115 linearly with phytoplankton concentration, $g_{H_I}(p) = g_0 p$. The saturating functional responses are linear at low prey concentration and saturate at high prey concentrations. An example saturating response is the Holling type II functional response, $g_{H_{II}}(p) = g_0 \frac{p}{p_0+p}$. This functional response assumes that processing of food and searching for food are mutually exclusive behaviors (*Visser*, 2007; *Kiørboe et al.*, 2018). The parameter $g_0$ is a function of processing time and the parameter $p_0$ is a function of both search and processing time. This parsimonious theoretical basis and ability to fit the parameters from ex- 120 perimental data has made this functional response one of the most commonly used (*Verity*, 1991; *Kiørboe et al.*, 2018). The Holling type III functional response has a reduction in grazing at low prey concentration. One formulation is a sigmoidal function, $g_{H_{III}}(p) = g_0 \frac{p^2}{p_0^2+p^2}$ which can be approximated as quadratic in $p$ for low $p$ and asymptotes to a constant rate for high $p$. This type III functional response can be derived as a generalization of the type II response where the search time is a linear function of prey concentration. Effectively, there is a prey refuge at low concentration because it is more difficult for predators 125 to find each prey item. There are other possible mechanisms for a type III functional response, including a threshold response by predators (*Mullin et al.*, 1975; *Ohman*, 1984) and prey switching (*Vallina et al.*, 2014). To compare the functional responses, we formulate the zooplankton specific grazing rate as a function of phytoplankton concentration

$$g(p) = g_0 \frac{(p/p_0)^{k-1}}{1 + (p/p_0)^{k-1}}. \tag{6}$$

The exponent $k$ determines the degree of non-linearity of the functional response. The Holling type II functional response is 130 $k = 2$ and the Holling type III response is $k = 3$. The parameter $p_0$ is a half saturation constant. When $p = p_0$, the grazing is at half of the maximum rate $g(p_0) = \frac{g_0}{2}$.

## 2.1 Grazing linear in phytoplankton concentration for constant zooplankton concentration: Critical Depth Hypothesis

Phytoplankton are known to respond faster than zooplankton to environmental changes (*Fileman and Leakey*, 2005). The critical depth hypothesis first proposed by Sverdrup (1953) assumes that such an assumption applies to the rapid onset of the spring bloom and proposed to model the phytoplankton growth rate according to equation (2), but setting $g(p)z = \frac{g_0}{p_0}z_0 p$ with $z_0$ the constant zooplankton concentration before the bloom onset,

$$\frac{Dp}{Dt} = \left( \frac{1}{K_d H(t)}\mu_0 - d_p - g_0 z_0/p_0 \right) p. \tag{7}$$

*Sverdrup* (1953) focused on the time at the end of winter when the mixed layer starts shoaling in response to spring atmospheric conditions and thus could ignore the entrainment, i.e. $s^+ = 0$. Under these assumptions the mixed layer depth $H(t)$ is the only time dependent parameter which can determine whether the phytoplankton concentration is exponentially decaying (winter conditions) or exponentially increasing (spring bloom onset). This gave rise to the widely applied 'critical depth hypothesis' which states that phytoplankton accumulation starts when the mixed layer shoals beyond a critical depth,

$$H_c = \frac{\mu_0}{(d_p + g_0 z_0/p_0)K_d}. \tag{8}$$

While the critical depth hypothesis has become the most widely accepted framework to interpret the onset of spring blooms–but there are growing objections (*Behrenfeld*, 2010)–it is not very useful to make quantitative predictions. The criterion requires knowledge of the grazing rate at the end of winter before bloom onset, which is very difficult to measure. Sometimes this obstacle is overcome by assuming that $g_0 z_0 \ll d_p$, in which case the critical depth dependence on grazing can be ignored. However, the assumption is likely inappropriate for most blooms where grazing is a main source of mortality immediately prior to bloom formation (*Calbet and Landry*, 2004; *Irigoien et al.*, 2005). For example, assuming a typical attenuation coefficient of $K_d = 0.05 \, \mathrm{m}^{-1}$ in the winter North Atlantic (*Organelli et al.*, 2017; *Mignot et al.*, 2018), where bloom onset is often observed at a critical depth of around 200 m (as reported in *Siegel et al.* (2002)), the ratio of growth to mortality rate, $\frac{d_p + g_0 z_0/p_0}{\mu_0}$, is predicted to be close to be $0.1$. Mortality timescales of phytoplankton are believed to be longer than ten times their division rates implying that grazing, not mortality, dominates phytoplankton losses at bloom onset (*López-Sandoval et al.*, 2014). A theory of blooms must therefore include a predictions of the zooplankton concentrations and their grazing rates at the end of winter, if it is to make falsifiable predictions. Additionally, on seasonal timescales there is substantial variation in zooplankton concentrations so a theory that includes variable phytoplankton and zooplankton concentrations is necessary. The goal of the next two sections is to present two models of grazing with a focus on wintertime conditions.

## 2.2 Grazing linear at low phytoplankton concentration: $g(p) \sim p$

Consider first the saturating (type II) grazing function. In winter, prey concentrations are very low and this function is approximately linear $g_{H_{II}}(p) \approx \frac{g_0}{p_0}p$ (Figure 2b). During the wintertime, as the mixed layer deepens, water from below the mixed layer is entrained, decreasing the concentration of the phytoplankton and zooplankton ($s^+ > 0$) but not their standing stock

$(s^- = 0)$.

$$\frac{DP}{Dt} = \frac{1}{H}\left(\frac{\mu_0}{K_d} - \frac{g_0}{p_0}Z\right)P - d_pP$$
$$\frac{DZ}{Dt} = \frac{1}{H}\left(a\frac{g_0}{p_0}P - d_zZ\right)Z \tag{9}$$

Assuming the natural mortality of phytoplankton is negligibly small, the growth and grazing terms in the $P$ and $Z$ equations have the same dependence on mixed layer depth $H$ and thus any increase in $H$ does not reduce grazing any more than it reduces the growth of phytoplankton. Consider for example a population in equilibrium, i.e. $\frac{dP}{dt} = \frac{dZ}{dt} = 0$. The equilibrium populations are $Z^* \approx \frac{\mu_0 p_0}{K_d g_0}$ and $P^* = \frac{d_z p_0}{a g_0}Z^* = \frac{d_z \mu_0 p_0^2}{K_d a g_0^2}$ are independent of $H$, and thus an equilibrium population will remain in equilibrium even as the mixed layer deepens (Figure 2a). If phytoplankton biomass decreases at some point in winter then

subsequent changes in mixed layer depth cannot trigger any biomass accumulation as long as the biological parameters $\mu_0$, $a$, $g_0$, $d_p$, $d_z$, and $K_d$ remain constant.

It could be rebutted that winter accumulation is possible if zooplankton mortality is represented as a linear, rather than quadratic loss term. In that case, as the mixed layer deepens, zooplankton biomass loss rates would not decrease as quickly as the rate of zooplankton grazing on phytoplankton, eventually reaching a crossing over point at which there would be a net loss

of zooplankton biomass and consequently an increase in phytoplankton biomass. This is the case of Lotka-Volterra predatory-prey dynamics in a variable environment (*Yorke and Anderson Jr*, 1973; *Dubois*, 1975). However, this model is problematic because a linear zooplankton mortality at low concentrations is only defensible in the absence of grazing by higher trophic levels. Such grazing is what is implicitly modelled with a quadratic mortality term such as the one used in equation (9).

### 2.3 Grazing quadratic at low phytoplankton concentration: $g(p) \sim p^2$

The situation is different if we prescribe a phytoplankton grazing function that decreases more rapidly than linearly as $p$ decreases. The Holling type III functional response is a popular choice and can be written as $g_{H_{III}}(p) = g_0\frac{p^2}{p^2+p_0^2}$, which can be approximated as $g_{H_{III}}(p) \approx \frac{g_0}{p_0^2}p^2$ at low prey concentration and asymptotically approaches a constant value for high $p$. With this functional response, the rate of change of biomass at low $p$ is given by,

$$\frac{DP}{Dt} = \frac{1}{H}\left(\frac{\mu_0}{K_d}P - \frac{g_0}{Hp_0^2}ZP^2\right) - d_pP$$
$$\frac{DZ}{Dt} = \frac{1}{H}\left(a\frac{g_0}{Hp_0^2}ZP^2 - d_zZ^2\right). \tag{10}$$

In this case, the grazing rate decreases faster than the phytoplankton growth rate as the mixed layer deepens due to the additional $\frac{1}{H}$ factor in the grazing term (Figure 2c). This opens the possibility of a net increase in phytoplankton biomass due to deepening of the mixed layer, consistent with the disturbance-recovery hypothesis (*Behrenfeld*, 2010; *Behrenfeld and Boss*, 2014). This is the key result of this paper. In what follows, we will use observations to explore the implications of this insight beyond the low phytoplankton limit.

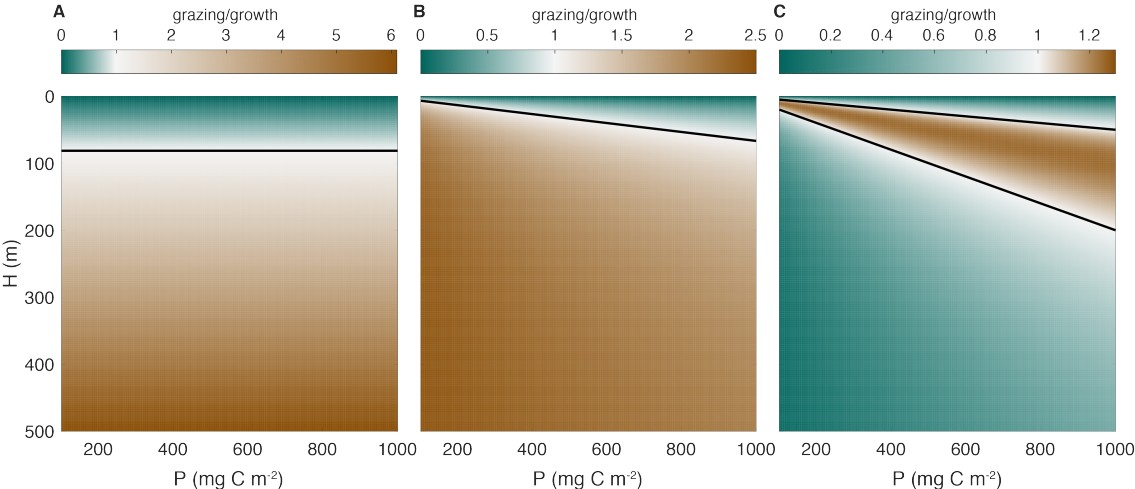

**Figure 2.** Ratio of grazing to growth as a function of phytoplankton biomass and mixed layer depth for mixed layer integrated models, e.g. equations 4. The black line separates regions where growth is faster than grazing (greens) from regions where grazing is faster than growth (browns). Note the different color scales in each panel. The parameter values used and the expressions for each functional type are given in table 1. (A) Holling type I (linear). (B) Holling type II (saturating). (C) Holling type III (inflection at low concentrations). The zooplankton concentration ($z_0$) used in panel A is $0.5$ mg C m$^{-3}$ and the zooplankton biomass ($Z$) used in panels B and C is $100$ mg C m$^{-2}$. It is well-established that growth dominates over grazing when the mixed layer is shallow due to increased light availability. In the case of the Holling type I functional response (A), the black line is the critical depth, which is independent of phytoplankton biomass. Note that if a constant value of zooplankton biomass rather than concentration is used, the ratio of growth to grazing using the Holling type I functional response is constant as in equation 9. The switchover between growth to grazing dominance depends on phytoplankton concentration in the case of a Holling type II functional response and therefore on the combination of mixed layer depth and phytoplankton biomass. Only in the case of Holling type III functional response is there also a decrease in the grazing rate as the mixed layer depth increases.

## 3  Modeling the annual cycle

We aim to demonstrate that when implemented in a full NPZ model, a grazing function with a quadratic (or higher) dependence on phytoplankton concentration at low $p$ is sufficient to reproduce both wintertime biomass accumulation and a spring bloom. In order to model the full annual cycle, we utilize a more realistic phytoplankton growth rate that depends on nutrient concentration and has a yearly cycle. We replace the growth term $\frac{\mu_0}{K_d H(t)}$ in equation (2) with

$$\mu(t,n) = \mu_0 \frac{n(t)}{n_0 + n(t)} \frac{I(t)}{I(t) + I_0} \frac{1}{K_d H(t)} \left(1 - e^{-K_d H(t)}\right). \tag{11}$$

where

$$I(t) = 20 \left(0.6 \sin\left(\frac{t + 270}{365} 2\pi\right) + 1\right) \tag{12}$$

This growth rate has temporal dependence through the mixed layer depth and through the surface irradiance. It also depends on nutrient concentration through the function $n/(n + n_0)$ which varies throughout the year and is close to one in winter when $p$

and $z$ are small. Although one could add other processes, this model can reasonably reproduce the seasonal cycle and illustrate our point. However, other processes may be need to represent all aspects of the annual cycle. Using this model we now test the impact of the grazing function on the yearly evolution of biomass and compare with in-situ observations.

**Float measurements of the phytoplankton annual cycle in the North Atlantic**

We calibrate the NPZ model using the averaged annual cycle of phytoplankton biomass as observed by BGC-Argo floats in
the high-latitude North Atlantic (*Mignot et al.*, 2018). Our model ignores the effect of lateral heterogeneity or restratification on phytoplankton dynamics (*Mahadevan et al.*, 2012; *Karimpour et al.*, 2018). In order to relate the model results to empirical data, we followed *Mignot et al.* (2018) and selected observations where vertical mixing dominates over lateral transport, i.e. trajectories where lateral density gradients that drive horizontal flows are weak. This was done by restricting the analysis to floats that did not cross into different water masses (defined as a change in water mass properties in T-S space). Twelve
annual cycles that met this criterion were observed during the period 2013-2016 between the latitudes of 50°N and 65°N. All individual float trajectories are plotted in the appendix of *Mignot et al.* (2018).

We estimated phytoplankton concentration $p(t)$ from backscatter measurements. The mixed layer depth $H(t)$ is defined as the depth at which the potential density increases by 0.03 kg m$^{-3}$ from the potential density at 10 m (*Kara et al.*, 2000). As in *Mignot et al.* (2018), the net phytoplankton population accumulation rate was then calculated using the observed phytoplankton
concentration and mixed layer depth as

$$r_p = \frac{1}{P} \int_{-H}^{0} \frac{\partial p}{\partial t} d\zeta = \frac{1}{P} \left( \frac{\partial P}{\partial t} - p(-H) \frac{\partial H}{\partial t} \right). \tag{13}$$

In contrast to *Mignot et al.* (2018), the accumulation rate was computed over the mixed layer rather than the productive layer. In order to account for interannual and regional variability in bloom timing, we rescaled the time axis of each individual float time series to account for variability in the start and end dates of winter and spring each year. The rescaled time is defined
as $\tau = \frac{t-t_1}{t_2-t_1}$ where $t_1$ is the calendar day of the onset of weak winter accumulation (the first time in the year when the accumulation rate is positive for at least 24 consecutive days) and $t_2$ is the calendar day of the onset of spring (the first time in the year when the mixed layer shoals for at least 24 consecutive days) (*Mignot et al.*, 2018). The average population growth rates was then estimated by averaging over all float time series as a function of the time $\tau$. The result is then plotted in Figure 3 as a function of calendar days setting $\tau = 0$ as the median of all $t_1$ and $\tau = 1$ as the median of all $t_2$.

**Model parameters**

The NPZ model equations (2) are solved replacing $\mu_0$ with $\mu(n,t)$ as given in equation (11) and using the yearly timeseries of $H(t)$ estimated from the average from all float measurements. The nitrogen concentration below the mixed layer is set to $N_{max} = 30$ mg N m$^{-3}$ based on the nitrate concentration observed at depth by the biogeochemical Argo floats. Phytoplankton and zooplankton are assumed to immediately remineralize once they die so that $n + p + z =$ constant when there is no en-
trainment or detrainment. Some parameters are prescribed based on reasonably well established values found in the literature:

$\mu_0$ = 0.8 day$^{-1}$ (*Eppley*, 1972; *Geider et al.*, 1998; *Bissinger et al.*, 2008), $a$=0.5 (*Landry et al.*, 1984; *Moore et al.*, 2001), $n_0$=4 mg N m$^{-3}$ (*Moore et al.*, 2001), $K_d$=0.05 m$^{-1}$, and $I_0$=40 $\mu$mol quanta m$^{-2}$ s$^{-1}$ (*Bouman et al.*, 2018). However, other parameters relating to grazing and zooplankton and phytoplankton mortality are more uncertain (see Table 1). The focus of this manuscript is on the functional formulation of the model. If the model cannot reproduce the key features of the observa-
tions for any values of the parameters, then the model must be rejected. If we can find parameter values for which the model reproduces key features of the observations, we then assess if those values are consistent with observational estimates. The parameters related to grazing and mortality are therefore calibrated by fitting each model accumulation rate and concentrations to observations over the full annual cycle. We use a trust-region-reflective least-squares algorithm (*Coleman and Li*, 1996). Prior values for the biological parameters were chosen based on estimates from the literature (*Moore et al.*, 2001; *Behrenfeld and*
*Boss*, 2014). Parameter values are constrained to remain within realistic bounds during fitting. We tested the sensitivity of our estimates to the priors by systematically varying the initial parameter choice within the range of values reported in empirical studies. While the fitting algorithm found multiple local minima, all the biologically sensible ones cluster around the values given in Table 1. The accumulation rates are smoothed before fitting with a five-point Savitzky-Golay filter. 84 data points are used in the fitting. The best fit parameters values are given in Table 1. Phytoplankton biomass is compared to the observations
in carbon units and conversions between nitrogen and carbon units are performed using a Redfield ratio of 16:106.

**Table 1.** Parameters used in figures 2, 3, and 4. Parameters above the line were prescribed based on literature values. Parameters below the line were fit by linear least squares parameter fitting of the phytoplankton growth rates. The final section lists the expressions used in figure 2. The integrated growth rate used in the ratio of grazing to growth is $\frac{\mu_0}{K_d}\frac{P}{H}$.

| Parameter | Significance (units) | Type I | Type II | Type III |
|---|---|---|---|---|
| $u_0$ | maximum growth rate (day$^{-1}$) | 0.8 | 0.8 | 0.8 |
| $a$ | zooplankton assimilation efficiency | 0.5 | 0.5 | 0.5 |
| $n_0$ | nutrient half saturation constant (mg N m$^{-3}$) | 4 | 4 | 4 |
| $N_{max}$ | deep nutrient concentration (mg N m$^{-3}$) | 30 | 30 | 30 |
| $K_d$ | attenuation coefficient (m$^{-1}$) | 0.05 | 0.05 | 0.05 |
| $I_0$ | light dependence ($\mu$mol quanta m$^{-2}$ s$^{-1}$) | 40 | 40 | 40 |
| $g_0$ | maximum grazing rate (day$^{-1}$) | - | 5.9 | 4.0 |
| $p_0$ | grazing saturation factor (mg C m$^{-3}$) | - | 15 | 15 |
| $d_z$ | zooplankton mortality rate (day$^{-1}$mg C$^{-1}$ m$^3$) | - | 6.0 | 1.8 |
| $d_p$ | phytoplankton linear mortality rate (day$^{-1}$) | - | 0.0004 | 0.001 |
| **Expression** | **Significance (units)** | **Type I** | **Type II** | **Type III** |
| G | integrated grazing rate (mg C m$^{-2}$ day$^{-1}$) | $\frac{g_0}{p_0}z_0\frac{P}{H}H$ | $g_0\frac{P/H}{P/H+p_0}\frac{Z}{H}H$ | $g_0\frac{(P/H)^2}{(P/H)^2+p_0^2}\frac{Z}{H}H$ |
| G/$\mu$ | ratio of grazing to growth | $\frac{g_0z_0K_d}{p_0\mu_0}H$ | $\frac{g_0K_d}{\mu_0}\frac{ZH}{P+Hp_0}$ | $\frac{g_0K_d}{\mu_0}\frac{ZPH}{P^2+H^2p_0^2}$ |

The temporal rescaling used to average the observational timeseries creates a spurious peak in net population growth rate at the beginning of winter (days 315 to 4). Throughout the winter there is variability in accumulation rates among individual

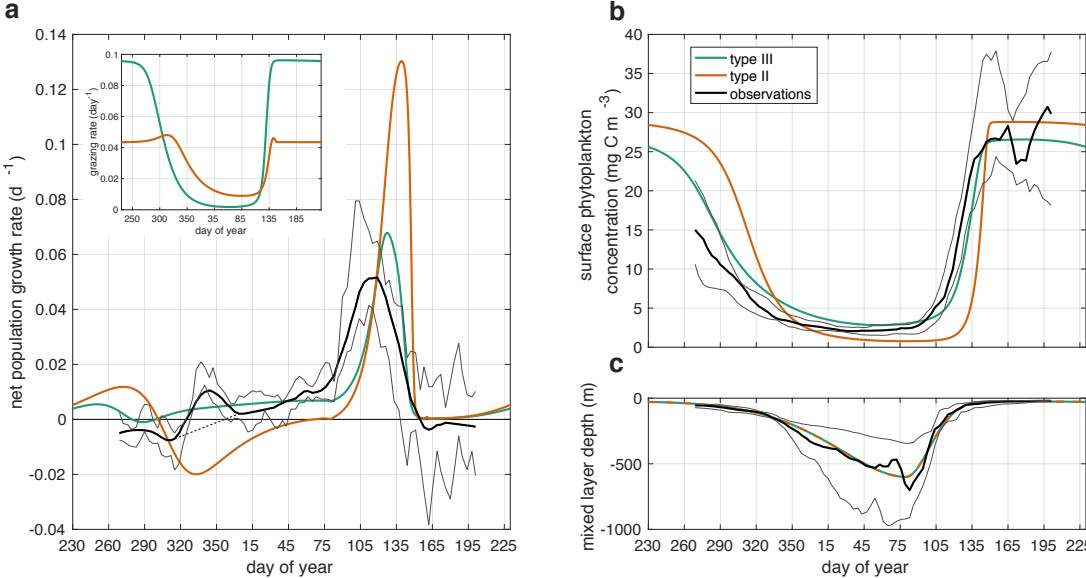

**Figure 3.** (a) Net mixed layer population growth rate in observations and model. Inset is grazing rate $g(p)$. The thin dashed black line from day 315 to day 4 shows the interpolated growth rate used in parameter fitting. (b) Annual cycle of phytoplankton surface concentration in observations and model simulations (c) Mixed layer depth. The observations, black line, are the median quantity measured by Argo floats with the interquartile range shown in grey lines. The green line is the prediction of the model in equations 1 with a Holling type III functional response. The orange line is the prediction of the model with a Holling type II functional response.

timeseries, including some negative values, even when the average is positive. Our choice to define the start of winter as the period when all timeseries have positive accumulation rates creates the spurious maximum in the observations at that time. We

remove this artifact before parameter fitting by interpolating linearly from day 315 to day 4 (Figure 3a).

### Comparison of model and observations

Using either Holling type II or III grazing functions, the model with the best fit parameters generates a spring bloom with a rapid increase in phytoplankton concentration and biomass that coincides with the spring shoaling of the mixed layer (Figure 3b). However, the Holling type III model results in net positive phytoplankton population growth through the winter, while

the Holling type II model does not (Figure 3a). The commonly used grazing functions of Holling type I and II do not satisfy the requirement of superlinear dependence of grazing on phytoplankton at low phytoplankton concentrations, and thus cannot capture the observed wintertime biomass accumulation, while the Holling type III functional response has the appropriate nonlinear dependence (*Holling*, 1959).

    During the winter (day 320-365 and continuing 1-75), the phytoplankton concentration is larger when using the type III

grazing function than when using the type II grazing function (Figure 3b). However, the winter grazing rate is lower with

the type III grazing function (Figure 3a). In order to compensate for the larger winter grazing with the type II function, the parameter fitting procedure infers a much lower linear phytoplankton mortality $d_p$ for that case (Table 1). One process that is included in the linear mortality in both cases is phytoplankton respiration. The linear mortality estimates from parameter fitting fall within the range of phytoplankton respiration rates from in situ observations and incubation experiments (*López-Sandoval et al.*, 2014; *Briggs et al.*, 2018).

During the summertime, the mixed layer depth is fairly constant and phytoplankton and zooplankton populations are close to equilibrium. This model does not include export from the mixed layer through sinking or migrating particles. Instead, any carbon export from the mixed layer only occurs when the mixed layer is shoaling due to biomass being left in the stratified layer below the new mixed layer.

The modeled relationship between phytoplankton and zooplankton shows notable differences between the two grazing functions. This is best illustrated by plotting the temporal evolution of the two communities in a $z - p$ plane as shown in Figure 4. At the end of winter, zooplankton are at slightly higher concentration with type III than type II grazing, because they have fared better throughout the winter by feeding on a larger phytoplankton population. Zooplankton respond slowly to the explosive spring phytoplankton bloom with the type II grazing, resulting in higher phytoplankton growth rates and a lower zooplankton concentration. By contrast, with the type III grazing, zooplankton are strongly coupled to phytoplankton and start grazing as soon as the bloom gets going reducing its amplitude. Importantly the rate of increase of phytoplankton concentration during the spring bloom is slower than exponential (*Mignot et al.*, 2018), consistent with the prediction of the disturbance-recovery hypothesis (*Behrenfeld and Boss*, 2014). With both grazing functions, the spring bloom populations are out of equilibrium with phytoplankton concentrations being higher and zooplankton concentrations being lower than at equilibrium.

The simple $n - p - z$ model used here is an imperfect representation of the observations. For example, the model only includes one phytoplankton type and one zooplankton type, which precludes both the succession of different phytoplankton types during the spring and summer and the presence of a microbial loop that could reduce the flow of carbon up the food chain (*Azam et al.*, 1983). The bulk zero-dimensional model assumes that phytoplankton and zooplankton concentrations are uniform in the mixed layer and zero below, a defensible approximation for winter conditions–the focus of this study–when the mixed layer is deep and turbulent mixing is strong, but not in other seasons when turbulence is weak (*Taylor and Ferrari*, 2011). The deficiencies of the bulk model are evident at the spring bloom onset–the time of dramatic acceleration of phytoplankton growth at the end of winter–which is slightly delayed in the model relative to the observations, occurring once the mixed layer has shoaled rather than during mixed layer shoaling. In observations, blooms start as soon as turbulent mixing subsides because phytoplankton is no longer mixed away from the surface, while there is a lag of days to weeks before the mixed layer restratifies and shoals (*Taylor and Ferrari*, 2011). The bulk model is also problematic in summer when the mixed layer is shallower than the euphotic layer and some of the productivity takes place below the mixed layer base where the model assumes $p = z = 0$. Despite these deficiencies, bulk mixed layer models have been shown to qualitatively reproduce the full annual cycle of plankton dynamics in other regions (c.f. *Evans and Parslow* (1985)) and are especially appropriate for our work which focuses on phytoplankton growth in winter.

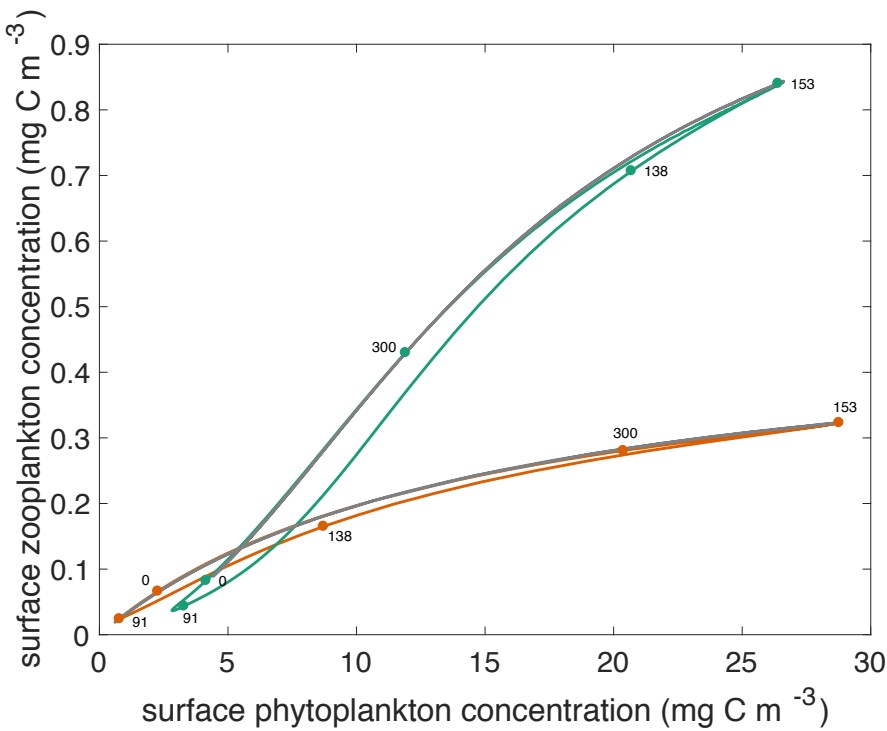

**Figure 4.** Surface phytoplankton and zooplankton concentrations through the annual cycle for both models. Green curve shows concentrations that result from the type III model. Orange curve shows concentrations that result from the type II model. The labeled dots indicate day of year for particular locations in the phase portrait. Over the annual cycle, the phytoplankton and zooplankton populations transit these curves counter-clockwise. The grey curves show the steady state concentrations throughout the annual cycle.

## 4 Discussion

Our work suggests that the winter accumulation of biomass recently documented from float observations in the North Atlantic (*Behrenfeld*, 2010; *Mignot et al.*, 2018), while much weaker than that the spring and summer accumulation (*Lutz et al.*, 2007; *Uitz et al.*, 2010), reveals otherwise hard to document top-down controls on phytoplankton populations. By studying winter time phytoplankton population dynamics, when growth conditions are less than optimal, we have been able to make inferences about the rate of zooplankton grazing. We demonstrated that the grazing rate as a function of phytoplankton concentration must decrease superlinearly at low phytoplankton concentrations in order to release the phytoplankton from grazing pressure. A quadratic grazer response function at low phytoplankton biomass is sufficient for phytoplankton biomass accumulation, although higher order nonlinearities would also reproduce the observed dynamics.

Relatively little is known about phytoplankton loss through grazing (*Dolan and McKeon*, 2005) in comparison to the other factors that control the dynamics of phytoplankton populations like macro- and micronutrients, light availability, and temperature (*Eppley*, 1972). Our work suggests that winter conditions may offer a unique opportunity to study phytoplankton grazing

in the field. In the wintertime, cell division rates decrease because of light limitation due to both deepening of the mixed layer and decrease in sea surface light. In order for phytoplankton accumulation rates to be positive while cell division rates are declining, the phytoplankton loss rates must decrease faster than division rates. While grazing is not the only concentration-dependent process, it is an interesting and compelling example of one process that could lead to biomass accumulation during the wintertime and it is the fundamental tenet of the "disturbance-recovery hypothesis" (*Behrenfeld and Boss*, 2018).

The wintertime growth is not only important to sustain phytoplankton populations in winter, but it is believed to play a crucial role for the development of the subsequent spring bloom. We showed that the reduction in grazing rate results in larger populations of both zooplankton and phytoplankton at the end of winter than would occur with a linear coupling. Furthermore, the larger zooplankton concentrations result in a faster acceleration in zooplankton grazing once phytoplankton concentrations increase during a bloom. The combination of more abundant wintertime populations and stronger/more rapid coupling between phytoplankton and zooplankton populations curb explosive phytoplankton growth. The magnitude and timing of the spring bloom and interactions between zooplankton and phytoplankton populations in the springtime may be affected by factors not considered here, such as a non-linearities in phytoplankton photophysiology, but our goal was to illustrate in as simple a framework as possible the impact of the functional form of grazing on winter growth and not to derive the most comprehensive model of phytoplankton phenology.

There are other possible explanations for wintertime biomass accumulation beyond the dilution of phytoplankton. The biological functions encapsulated in the NPZ model parameters may vary over time. For example, the zooplankton assimilation rate, $a$, could depend on the nutrient content of the prey introducing an alternative nonlinear effect (*Landry et al.*, 1984; *Irigoien et al.*, 2005). In our model, the only time-dependent terms are maximum insolation, which has little influence on wintertime biomass accumulation, and mixed layer depth, which drives the wintertime biomass accumulation. Additional factors such as temperature, which is correlated with mixed layer depth, may also have an impact on wintertime growth and grazing, representing another possible non-linear effect (*Rose and Caron*, 2007; *López-Urrutia*, 2008; *Chen et al.*, 2012). Functional diversity beyond that included in this model is also likely important. For example, mixotrophic metabolisms may contribute to phytoplankton accumulation in light-limited conditions (*Barton et al.*, 2013; *Flynn et al.*, 2013; *Leles et al.*, 2020). Finally, there is evidence that wintertime growth can be triggered by mixed layer instabilities that occasionally restratify the mixed layer during the winter and thus increase the light available for phytoplankton (*Taylor and Ferrari*, 2011; *Karimpour et al.*, 2018). However this cannot be the unique explanation, because float observations presented in (*Mignot et al.*, 2018) and reviewed here show many examples of wintertime accumulation where these mixed layer dynamics did not seem to apply.

The sensitive dependence of phytoplankton phenology on the rate of grazing by higher trophic levels at low concentrations provides a powerful quantitative framework in which to evaluate theories of plankton phenology. Observations of wintertime phytoplankton biomass accumulation have been interpreted as evidence of a release from grazing pressure in deep mixed layers, but little attention has been given to the key role played by the choice of grazing functions in these theories. Some studies have used a Holling type III grazing function (*Behrenfeld and Boss*, 2014; *Yang et al.*, 2020) while others relied on a prey switching formulation, where the zooplankton preferentially consumes the most common type of phytoplankton (*Llort et al.*, 2015). The observational evidence for a lower bound on phytoplankton concentrations (*Lessard and Murrell*, 1998) ought

to be studied within the framework presented here. The interpretation of the response of phytoplankton to sudden environmental perturbations on subseasonal timescales, such as storms (*Behrenfeld and Boss*, 2018), will also require a careful assessment of the grazing functions which control how fast zooplankton grazing responds to increases in phytoplankton concentrations. Last, but not least, this framework ought to be applied to test the predictions of different theories of bloom onset (*Verity et al.*, 1993; *Morison et al.*, 2020; *Mojica et al.*, 2020). While our work pointed out the key role played by the choice of grazing functions in such theories and models, it is important to note that state of the art Earth System Models often use multi-species ecosystem models (*Laufkötter et al.*, 2015). Multispecies models do not necessarily result in the same dynamics as the single-species functional responses used here (*Gentleman et al.*, 2003), but our result that phytoplankton phenology is very sensitive to the degree of non-linearity in growth and mortality functions is very likely to hold for more complex ecosystem models as well.

It is worth commenting on the ecological underpinnings for the different models of grazing. The grazing functions used in our model represent the coupling between all species of each trophic level of phytoplankton and zooplankton; the phytoplankton class includes all autotrophs, while the zooplankton class includes all grazers that consume phytoplankton. A superlinear decrease in grazing rates at low prey concentration has been observed in the lab studies of aquatic vertebrates and invertebrates and in theoretical studies (*Real*, 1977; *Barrios-O'Neill et al.*, 2016). In the China Seas, the microzooplankton grazing rates are best described by a Holling type III functional response (*Liu et al.*, 2021), providing evidence for the applicability of this functional response to whole populations, at least in the low and mid latitudes. Similarly, copepods go into diapause in the wintertime (*Baumgartner and Tarrant*, 2017), and this effectively reduces grazing pressure during winter; however, microzooplankton account for the majority of phytoplankton mortality in the ocean (*Landry and Calbet*, 2004). Other mechanisms described by the Holling type III functional response include prey switching (*Vallina et al.*, 2014), predator learning (*Holling*, 1966), and prey refuges (*Taylor*, 2013). While there are few structural prey refuges in an oceanic mixed layer, a patchy environment can also provide a type of prey refuge. A Holling type III functional response can arise due to non-random grazing behavior when population dynamics are integrated over a patchy environment (*Nachman*, 2006; *Morozov*, 2010).

## 5 Conclusions

A reduction in the grazing rate at low phytoplankton concentration has been proposed as the mechanism to explain the emerging observation that biomass often increases, albeit weakly, during the wintertime when mixed layers deepen (*Behrenfeld*, 2010). It has also been pointed out that the critical depth hypothesis fails to capture wintertime growth because it implicitly assumes that loss rates are constant either because they are dominated by constant respiration or by grazing by a constant zooplankton population. Previous modeling results have not acknowledged that a reduction in grazing pressure through dilution of plankton populations in deep mixed layers requires a grazing function that decreases faster than linearly in phytoplankton concentration at low concentrations.

While our analysis focused on wintertime conditions, we believe that more attention on the functional form of grazing functions may shed light on other phases of phytoplankton phenology as well. Observations show a tight coupling between the evolution of phytoplankton and zooplankton populations in all seasons *Stelfox-Widdicombe et al.* (2000); *Karayanni et al.*

375     (2005). This coupling has been interpreted as evidence that zooplankton grazing pressure can respond very rapidly to any changes in phytoplankton concentrations (*Behrenfeld and Boss*, 2018). To the extent that these interpretations are correct, our study suggests that observations can therefore be used to infer the functional form of grazing functions, an aspect of plankton ecology that is otherwise very difficult to quantify.

    Observational validation of the functional forms of grazing functions is key to build confidence in predictions based on
380   biogeochemical models. Different models can be tuned to provide reasonable estimates of the annual cycle of phytoplankton biomass, like our NPZ model with a saturating grazing function. However, in order to make predictions that are robust to changing conditions, it is important that models have the correct functional dependencies. Tuning of model parameters is no guarantee of model performance in an evolving environment that has not been observed yet. Climate change may reshape North Atlantic phytoplankton populations and primary production (*Balaguru et al.*, 2018) due to increasing surface tempera-
385  ture, shoaling mixed layer depths, and increasing upper ocean stratification (*Edwards and Richardson*, 2004). Predicting and quantifying the impact of these changes requires robust model formulations, not models tuned to present climate conditions.

*Code and data availability.*   All code and data are available at https://github.com/mara-freilich/grazing_functions_bg (DOI: 10.5281/zenodo.4282657)

*Author contributions.*   All authors conceptualized the research. AM curated the BGC Argo data. MAF wrote the model code and performed
390  the simulations and model analysis. MAF, GF, and RF performed the mathematical analysis. MAF wrote the manuscript with input from all co-authors.

*Competing interests.*   The authors declare that they have no conflict of interest.

*Acknowledgements.*   The authors would like to acknowledge funding from an NDSEG fellowship and Martin fellowship to MAF. The authors would like to thank Amala Mahadevan, Stephanie Dutkiewicz, Emmanuel Boss, and three anonymous reviewers for feedback on earlier drafts
395  of this manuscript.

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
