# Peer review of "Grazing behavior and winter phytoplankton accumulation"

_Biogeosciences, 2020_

## Referee Comment (RC1) · Anonymous Referee #1 · 15 Dec 2020

The manuscript deals with the mathematical description of grazing behaviors which may explain the (weak) increase in phytoplankton biomass observed in winter in the North Atlantic Ocean. The positive growth of phytoplankton during winter, light limiting conditions was previously explained by invoking a decrease in grazing pressure. More specifically, it was suggested that, due to the dilution of the water column caused be the deepening of the mixed layer, the encounter rate between phytoplankton and their grazers decreases, and this allows a weak positive growth despite light limitation. The authors highlight that this behavior can be modelled assuming a non-linear (in this case quadratic) relationship between grazing and prey biomass, a functional response known as Holling type III function. The main conclusion of the work is that assuming

this latter functional response a simple biogeochemical model is able to simulate both the increase in biomass in winter and the subsequent spring bloom, while other widely used grazing functional responses underestimate winter phytoplankton biomass. The Holling type III model is not new and is, as also stated by the authors, already a quite popular choice within the marine modelling community. However, this paper provides a remarkably clear explanation on why this formulation should be preferred over the others, also highlighting its ecological significance. The manuscript is clearly written, model's assumptions are very well explained and the results are clear and logic. I recommend publishing this paper after minor revisions.

Specific comments: I appreciate that this paper focuses on zooplankton. However, other phytoplankton loss terms can be equally important in the formation and progression of a blooms (including relatively high winter biomass). For example, viral infection, possibly one of the main causes of mortality after grazing, could also be described by a non-linear, density dependent functional response (e.g. Mateus, 2017, FMS), and have a reduced effect on phytoplankton at low biomass concentration.

The presented model follows the classic dichotomy between autotrophic and heterotrophic organisms. However, we know that most phytoplankton exhibits mixotrophic metabolism, perhaps with the important exception of diatoms (see e.g. Flynn et al 2012 JPR, Gonçalves Leles et al., 2018 JPR and 2020 Progress in oceanography). I am wondering if mixotrophy could be involved in the increase of biomass observed under light limiting conditions.

Figure1 is a bit misleading. I think that the different functional responses (II and III) need to be run with the same parameters in order to clearly assess differences and similarities.

Minor points: Line 88: the term $1/k_dH(t)$ is "the average light over the mixed layer.." but light is not explicitly modelled, right? Is the specific growth rate which is averaged over the ML, assuming an exponential decay. Line 110: the formulation (type III) "is

quadratic in p for low p". Why only for low p? I see that the focus here is on low winter biomass values but the formulation is quadratic for any p.

---

## Referee Comment (RC2) · Anonymous Referee #2 · 5 Jan 2021

**1   General comments**

This manuscript aims at explaining the phytoplankton growth rate observed during winter in the ocean mixed layer, when conditions are usually not conducive to biomass accumulation. The work is stimulated by the recent availability of data offered by the BGC-Argo platforms, which are providing unprecedented information on the open ocean microbiome. Positive winter growth rates have been recently identified using Argo floats under sea ice, both in the Arctic and in the Antarctic (Randelhoff et al., 2020; Hague and Vichi, 2020). This is an indication that, even in extreme conditions of light limitations (and likely with minimal predation pressure), phytoplankton have developed the capacity to grow and accumulate biomass. It is therefore (nowadays) less surprising that this may occur in the winter mixed layer of the North Atlantic Ocean. It is clearly a balance between growth and loss terms, and we are only now - thanks to the new autonomous technologies - starting to understand some of the details of the phytoplanktonic component.

This work is however timely and of relevance for the discussion, but I have some issues with the current manuscript, some of them that I consider quite relevant in order to gain confidence in the presented results. In its current form, I would recommend a thorough revision of the introduction, assumptions and the related methodologies, to ensure that the results are of wider applicability.

1. My first major comment is related to the focus on the grazing formulation in the context of winter biomass accumulation. To emphasize the importance of grazing in tilting the balance of winter phytoplankton development, the authors state that phytoplankton loss is due primarily to grazing and cellular lysis, and secondarily to respiration and natural mortality. Notwithstanding the quite ambiguous definition of "natural mortality", which should very likely include cellular lysis induced by viruses, they attribute this sentence to Landry and Calbet (2004). I would argue that the cited paper, as well as the twin publication Calbet and Landry (2004), specifically meant to highlight the role of microzooplankton grazing in the microbial food chain, through the analysis of dilution experiments. It is not a generic study on the grazing relationship between phytoplankton and their predators, neither specifically targeting the winter period, which is the focus of this manuscript. I would suggest the authors present their case with more support from the existing literature. I would also recommend the authors to be more rigorous in their use of the concept of linear and nonlinear grazing terms, because I have found their terminology, as well as its use in the presentation of the results, confusing at times. The grazing term of the dynamics is always nonlinear (it is the product of the two state variables P and Z), while the formulation of the grazing rate can

be linear. Apart from some theoretical works in the 80's and early 90's (which can be found in the cited work by Gentleman et al, 2003), there is not one single "state-of-the-art" biogeochemical model that uses a linear formulation for the specific grazing rate (at least to my knowledge. I checked PISCES, BFM, BioE-BUS, MEDUSA and a few others and they implement various combinations of hyperbolic functions or Ivlev's. I'm afraid the current wording may mislead the reader to assume that the majority of models uses a linear formulation, and this is why they may fail to show the winter accumulation. Throughout the manuscript I started to realize that maybe the authors intended that the hyperbolic grazing rate function is linear at "low prey concentrations". I concede that this is true for biomass values close to 0, as can be seen with a Taylor series expansion of the term around 0. For any value larger than 0, it however depends on the choice of the grazing saturation factor. My major issue with Sec. 2 results is that the authors made the arbitrary assumption that the Holling type II is equivalent to a linear dependence on phytoplankton concentration, which is only true in the approximation of $P \to 0$. (Incidentally, please note that it is not always clear which parameters the authors used in Sec. 2, so I would suggest to improve this description). Fig. 2 thus presents a comparison between the Holling type I and Holling type III formulations, without considering Holling type II. The resulting tuned value of parameter $p_0$ for the Holling type II formulation applied in Sec. 3 is rather large, which forces this grazing rate to be in the linear part of the curve (Fig. 1) for the lowest modelled values of phytoplankton. I am therefore not much confident about the robustness of the results shown in Sec. 3, as I suspect some degree of overfitting led to the observed difference between the two formulations. This comment is further elaborated in point 4 below.

2. Another major issue I have with this manuscript is the apparent naivety in the biogeochemistry formulation and the related terminology. With all due respect, the sentences in line 65 and line 209 would have made me reject the manuscript

immediately, since there is no carbon content in any dissolved inorganic nutrient and nitrogen pool cannot be described in units of carbon. I truly believe in the value of theoretical simplified models, and therefore I may interpret what the authors meant with these statements (this is why in mathematical modelling there is a tendency to use non-dimensional formulations). Nonetheless it is a major mistake that would confuse the non-modeller, and it is found throughout the manuscript. This actually invalidates any use of other nutrient-related parameters taken from the literature (because they are usually in units of nitrogen or with multiple "currencies"), unless the authors state in advance what is the stoichiometric relationship between nutrients and carbon in their compartments and adjust the values accordingly. For instance, they refer to the nitrate half-saturation constant used by Moore et al. (2001), but in their appendix A this value is in mmol/m3. I would recommend the authors to present their formulations using the appropriate units for the generic nutrient, eventually adopting the use of constant stoichiometric ratios if needed.

3. The authors make the statement that loss terms other than grazing are of lower orders, but I argued in point 1 above that this is a not fully justified statement. I would thus recommend to explain more in detail their choices for the model formulation. For instance, does phytoplankton mortality parameterize respiration and viral lysis? This is a rather important concept, since this parameter has been considered freely tunable (see my point 3 below). I would also argue that the authors have not given much attention to the choice of the zooplankton closure formulation. Is there a specific reason for not using a linear mortality as indicated by Edwards and Yool (2000)? This is briefly mentioned at line 155 but without context it is difficult to appreciate its role. The disturbance-recovery hypothesis would also affect zooplankton mortality rates, which I understand was the authors' interpretation since they treated this parameter as freely tunable. However, if this parameter is re-tuned when the mathematical term is changed, then it is difficult

to assess the outcome. In addition, the authors do not always explain the use of certain mathematical notations. For instance, they use material derivatives in eq. (1) but make no assumptions as to why they are used and eventually turned into partial derivatives in eq. (2).

4. Another major point is linked to assuming the phytoplankton mortality constant ($d_p$) dependent on the mathematical form of the grazing rate. If this loss term is meant to parameterise phytoplankton mortality other than grazing, should not this be treated as independent of zooplankton and hence belonging to the fixed set of parameters in Table 1? This is a first-order loss term, which is usually interpreted as basal (or biomass-related) metabolic losses. It may indeed also include viral lysis, although in this case it should be density-dependent (see the excellent review in Mateus, 2017). I would recommend the authors to repeat their experiments assuming a constant value for this specific loss term and simply attributing it to basal respiration, which is the more plausible metabolic rate. It is not completely clear to me if the parameters calibrated in Table 1 have been used for the theoretical analysis presented in Sec. 2 and Fig. 2. Edwards and Yool (2000) gave clear indication that the role of parameters is often more important than the choice of the mathematical formulations, and it should be demonstrated that it is not the case here. In relation to Sec. 3, what would be very important, is to ensure that the overwintering phytoplankton biomass is the same in both the experiments, and that all parameters but the ones related to grazing terms are fixed. I do not agree much with the statement that this is an unknown (and thus freely-tunable) term. They can find a range of values in Lopez-Sandoval et al. (2014; between 9 and 22At line 139 the authors state that mortality time scales are 10 times the scale of division rates. This is an unclear and unsupported statement (division rate can be made equivalent to a carbon fixation, or production, rate using some algebra, but I think the authors should compare like with like). The paper cited above reports it to be about 10

5. The dilution hypothesis is one of the possible explanations for winter accumulation of primary producer biomass, however this is based on the assumptions that light limitation is preclusive of any photon assimilation. Even if I did not expect the authors to be aware of the very recent papers on phytoplankton phenology cited above, I would have expected this work to give a more proper consideration to the role of light as a driver of phytoplankton growth. Not all the assumptions are justified, and particularly how the various processes have been averaged in the bulk mixed layer model. The authors added an implicit treatment of patchiness due to the entrainment of phytoplankton-depleted water from the deeper ocean, but ignored and did not discuss the inhomogeneity of growth due to the exponential decay of light in the mixed layer. This implies that the authors assume that the average phytoplankton growth is determined by the average light, thus assuming negligible that the Sverdrup himself suggested that growth may take place even with strong turbulence if the phytoplankton displays a positive phototaxis. This problem has been treated extensively in Paparella and Vichi (2020), both considering cases in which the biological rates are quicker than the physical scales and the opposite, which is the main founding assumption done by Sverdrup and also made by the authors. In both cases, it has

6. Finally, I'm afraid the authors gave a somewhat biased interpretation of the discussion about the Sverdrup model and instrumentally presented it to reinforce their argumentations. It is now widely accepted that the Sverdrup model is a conceptual mathematical formulation that helped to promote the quantitative study of the ocean microbiome, in an era when marine biogeochemistry was still a branch of descriptive biology. This is clearly illustrated in the cited paper by Fischer et al, and even more in the special issue of the ICES journal edited by Sathyendranath et al. (2015). Citing from the abstract, this set of papers celebrates an elegant and powerful hypothesis that has had long-lasting influence. Sverdrup clearly intended to describe the April-May period and not the winter accumulation, and

therefore it should not be used as an alternative hypothesis for the winter case. It is also important to remark that none of the current biogeochemical models make predictions based on the Sverdrup hypothesis. As such, this is more to be seen as an emergent property, rather than an explanatory mechanism. I have thus been quite surprised to see that the authors made their own revisitation of the Sverdrup model in Sec. 2.1, attributing their interpretation to the original formulation. Sverdrup introduced just a first-order loss term and then, eventually, made a few considerations about the role of grazing, showing that the data available at that time were in agreement with his postulation. It is the authors' interpretation to attribute this constant loss term to grazing, and it should be made very clear. I would particularly recommend removing it from the title of sec. 2.1 because it would be rather misleading to those readers who are less familiar with the original paper.

**2 Specific comments**

L143 Concentration L152 (and L155-157) A decrease in MLD would likely enhance growth terms more than the grazing losses (see point 5 above). Since episodes of re-stratification are actually frequent in the ocean (e.g. Smith et al., 2016), this assumption may not hold even during fully mixed conditions. L177 Please indicate which equation set is being used here L183 More details are needed to explain the modelling of the nutrient concentration and how this has been constrained according to observations. The use of a generic nutrient variable is sensible in Sec. 2, but here, because of the connection with the specific BGC-Argo measurements, it is important to clarify which nutrient is being used and how it has been assessed in terms of seasonal cycle. The floats do not measure nitrate, but other historical data can be used. L191 Biology has been demonstrated to be patchier than physics in the North Atlantic (e.g. Mahadevan and Campbell, 2002). Please consider this in your discussion. L192-193. There are

not many figures in this manuscript. I would thus suggest the authors to add some more information on the floats that have been used, and possibly provide a table with their identification numbers. It would be valuable to see the individual timeseries, to appreciate the phenological variability prior to their standardization. L199,L202, and others. I would recommend the authors to be consistent in their terminology. There is a tendency to consider the term accumulation and growth as synonyms, however, there can be accumulation without growth. L205 I would need some more explanation here. According to the description, I was expecting a non-dimensional axis and not calendar days. If the model was not rescaled, what are the t1 and t2 values for the model (and actually, having two different model formulations, which one was chosen?) L217 I understand that this is more of a personal choice, but I think there is value in introducing the methodology at the beginning of the paper. The parameter fitting procedure is now interrupting the presentation of the results. Also, the actual description of the optimization procedure is shown later at lines 224-227. Many of my major issues are related to the possible influence of overfitting that may have biased the results, and I think the authors should make an effort to demonstrate that this is not the case. I am not much familiar with this methodology and therefore it would be useful to know if the parameters have been tested one-at-a-time and what kind of sensitivity analysis was performed. L231-232. Please indicate if eq. (12) was used to estimate this rate. I struggle to understand why the rate is positive but concentration keeps decreasing. L273-276 My concerns on this statement are expressed in the major points above. The degree of nonlinearity of the Holling type II formulation depends on the value of the parameter, which is tuned differently in the two experiments, as well as the basal loss rates. The response of zooplankton would largely depend on the extant biomass during winter, which is quite different in the two simulations. L294-295 Light dependence was made less influential by assumption (see point 6 above). Please rephrase.

**3 References**

Calbet, A., Landry, M.R., 2004. Phytoplankton growth, microzooplankton grazing, and carbon cycling in marine systems. Limnology and Oceanography 49, 51–57. https://doi.org/10.4319/lo.2004.49.1.0051

Edwards, A.M., Yool, A., 2000. The role of higher predation in plankton population models. Journal of Plankton Research 22, 1085–1112. https://doi.org/10.1093/plankt/22.6.1085

Hague, M. and Vichi, M., 2020. Southern Ocean BGC-Argo Detect Under Ice Phytoplankton Growth Before Sea Ice Retreat, Biogeosciences Discuss. [preprint, accepted for publication in Biogeosciences], https://doi.org/10.5194/bg-2020-257

López-Sandoval, D.C., Rodríguez-Ramos, T., Cermeño, P., Sobrino, C., Marañón, E., 2014. Photosynthesis and respiration in marine phytoplankton: Relationship with cell size, taxonomic affiliation, and growth phase. Journal of Experimental Marine Biology and Ecology 457, 151–159. https://doi.org/10.1016/j.jembe.2014.04.013

Mahadevan, A., Campbell, J., 2002. Biogeochemical patchiness at the sea surface. Geophysical Research Letters 29, 1926.

May, R.M., 2004. Uses and Abuses of Mathematics in Biology. Science 303, 790–793. https://doi.org/10.1126/science.1094442

Mateus, M.D., 2017. Bridging the Gap between Knowing and Modeling Viruses in Marine Systems—An Upcoming Frontier. Front. Mar. Sci. 3. https://doi.org/10.3389/fmars.2016.00284

Paparella, F., Vichi, M., 2020. Stirring, Mixing, Growing: Microscale Processes Change Larger Scale Phytoplankton Dynamics. Front. Mar. Sci. 7. https://doi.org/10.3389/fmars.2020.00654

Randelhoff, A., Lacour, L., Marec, C., Leymarie, E., Lagunas, J., Xing, X., Darnis, G., Penkerc'h, C., Sampei, M., Fortier, L., D'Ortenzio, F., Claustre, H., Babin, M., 2020. Arctic mid-winter phytoplankton growth revealed by autonomous profilers. Science Advances 6, eabc2678. https://doi.org/10.1126/sciadv.abc2678

Sathyendranath, S., Ji, R., Browman, H.I., 2015. Revisiting Sverdrup's critical depth hypothesis. ICES J Mar Sci 72, 1892–1896. https://doi.org/10.1093/icesjms/fsv110

Smith, K.M., Hamlington, P.E., Fox-Kemper, B., 2016. Effects of submesoscale turbulence on ocean tracers. Journal of Geophysical Research: Oceans 121, 908–933. https://doi.org/10.1002/2015JC011089

———————————————

---

## Author Comment (AC1) · 6 Feb 2021

We thank the reviewer for their thoughtful comments. We agree with the reviewer about this manuscript's main contribution to the literature and appreciate the suggestions to improve the manuscript. We outline the changes we have made based on these comments below. We have included excerpts of the reviewer's comments in italics.

*I appreciate that this paper focuses on zooplankton. However, other phytoplankton loss terms can be equally important in the formation and progression of a blooms (including relatively high winter biomass). For example, viral infection, possibly one of the main causes of mortality after grazing, could also be described by a non-linear,*

[Figure]

*density dependent functional response (e.g. Mateus, 2017, FMS), and have a reduced effect on phytoplankton at low biomass concentration.*

As long as the mortality has a nonlinear dependence on P, our conclusions remain unchanged. It was indeed an oversight to neglect viruses in the discussion as they clearly contribute to phytoplankton mortality. Their inclusion would further strengthen our conclusions. We have added the following sentence in the presentation of the model "The grazing function represents a density-dependent mortality process. Other mortality processes such as viruses may also be density-dependent and could be studied in a similar manner but are not included in this analysis (Weitz et al 2015, Mateus 2017). Some of the results may nonetheless be applicable to other density-dependent mortality processes."

*The presented model follows the classic dichotomy between autotrophic and heterotrophic organisms. However, we know that most phytoplankton exhibits mixotrophic metabolism, perhaps with the important exception of diatoms (see e.g. Flynn et al 2012 JPR, Gonçalves Leles et al., 2018 JPR and 2020 Progress in oceanography). I am wondering if mixotrophy could be involved in the increase of biomass observed under light limiting conditions.*

We agree that mixotrophy could be involved in the increase of biomass during the wintertime. Further modeling and observational studies would be needed to evaluate the role of heterotrophy and mixotrophy in the wintertime North Atlantic. We have now included this point in the discussion by adding the following sentence on line 295: "Functional diversity beyond that included in this model is also likely important. For example, mixotrophic metabolisms may contribute to phytoplankton accumulation in light-limited conditions (Barton et al. 2013, Flynn et al. 2013, Gonçalves Leles et al. 2020)."

*Figure1 is a bit misleading. I think that the different functional responses (II and III) need to be run with the same parameters in order to clearly assess differences and*

*similarities.*

As it is currently presented Figure 1 aids with interpretation of Figures 3 and 4, which use the optimal parameters for each functional type. We need to use different parameters for each model otherwise the model for which the parameters are optimized will by definition perform better. We have now plotted the functional responses with the same parameters for both functional grazing functions in an appendix to aid in the comparison of the functional types.

*Line 88: the term 1/kdH(t) is "the average light over the mixed layer.." but light is not explicitly modelled, right? Is the specific growth rate which is averaged over the ML, assuming an exponential decay*

It is correct that light is not explicitly modeled. We have revised this sentence for clarity to read "The term $\mu_0/K_d H(t)$ is the average growth rate over the mixed layer, which is computed as the integral of the light-depth growth over the mixed layer depth divided by the mixed layer depth."

*Line 110: the formulation (type III) "is quadratic in p for low p". Why only for low p? I see that the focus here is on low winter biomass values but the formulation is quadratic for any p.*

We have modified this sentence for clarity. It now reads "is quadratic in p for low p and asymptotes to a constant rate at high p."

---

## Author Comment (AC2) · 6 Feb 2021

We thank the reviewer for their thorough and thoughtful comments. We have made revisions and clarifications where appropriate, as outlined below. We have included the reviewer comments in this response and respond in line. The reviewer comments are in italics.

*This manuscript aims at explaining the phytoplankton growth rate observed during winter in the ocean mixed layer, when conditions are usually not conducive to biomass accumulation. The work is stimulated by the recent availability of data offered by the BGC-Argo platforms, which are providing unprecedented information on the open*

*ocean microbiome. Positive winter growth rates have been recently identified using Argo floats under sea ice, both in the Arctic and in the Antarctic (Randelhoff et al., 2020; Hague and Vichi, 2020). This is an indication that, even in extreme conditions of light limitations (and likely with minimal predation pressure), phytoplankton have developed the capacity to grow and accumulate biomass. It is therefore (nowadays) less surprising that this may occur in the winter mixed layer of the North Atlantic Ocean. It is clearly a balance between growth and loss terms, and we are only now - thanks to the new autonomous technologies - starting to understand some of the details of the phytoplanktonic component. This work is however timely and of relevance for the discussion, but I have some issues with the current manuscript, some of them that I consider quite relevant in order to gain confidence in the presented results. In its current form, I would recommend a thorough revision of the introduction, assumptions and the related methodologies, to ensure that the results are of wider applicability.*

This manuscript focuses on the mechanisms underlying the wintertime biomass accumulation, which has previously been reported from autonomous observations. We have added the recent references to the introduction.

*1. My first major comment is related to the focus on the grazing formulation in the context of winter biomass accumulation. To emphasize the importance of grazing in tilting the balance of winter phytoplankton development, the authors state that phytoplankton loss is due primarily to grazing and cellular lysis, and secondarily to respiration and natural mortality. Notwithstanding the quite ambiguous definition of "natural mortality", which should very likely include cellular lysis induced by viruses, they attribute this sentence to Landry and Calbet (2004). I would argue that the cited paper, as well as the twin publication Calbet and Landry (2004), specifically meant to highlight the role of microzooplankton grazing in the microbial food chain, through the analysis of dilution experiments. It is not a generic study on the grazing relationship between phytoplankton and their predators, neither specifically targeting the winter period, which is the focus of this manuscript. I would suggest the authors present their case with more*

*support from the existing literature.*

We have rephrased this sentence to now read "Phytoplankton are thought to be tightly controlled by grazing and loss processes within the microbial food web (Landry and Calbet 2004, Calbet and Landry 2004, Strom et al 2007, Evans et al 2012, Prowe et al 2012). Any accumulation depends on the imbalance between growth and loss processes, the magnitudes of which are correlated to each other (Behrenfeld and Boss 2017)." We have also added additional citations to observational programs that observed that top down control is important, although given observational limitations at high latitudes that are addressed in the introduction, these are not wintertime observations. However, we have also included a citation to modeling work that also reveals the importance of top down control.

*I would also recommend the authors to be more rigorous in their use of the concept of linear and nonlinear grazing terms, because I have found their terminology, as well as its use in the presentation of the results, confusing at times. The grazing term of the dynamics is always nonlinear (it is the product of the two state variables P and Z), while the formulation of the grazing rate can be linear. Apart from some theoretical works in the 80's and early 90's (which can be found in the cited work by Gentleman et al, 2003), there is not one single "state-of-the-art" biogeochemical model that uses a linear formulation for the specific grazing rate (at least to my knowledge. I checked PISCES, BFM, BioEBUS, MEDUSA and a few others and they implement various combinations of hyperbolic functions or Ivlev's. I'm afraid the current wording may mislead the reader to assume that the majority of models uses a linear formulation, and this is why they may fail to show the winter accumulation. Throughout the manuscript I started to realize that maybe the authors intended that the hyperbolic grazing rate function is linear at "low prey concentrations". I concede that this is true for biomass values close to 0, as can be seen with a Taylor series expansion of the term around 0. For any value larger than 0, it however depends on the choice of the grazing saturation factor.*

The reviewer has made the valuable comment that the wording about the types of nonlinearity that can promote wintertime biomass accumulation should be more precise. We have edited the manuscript text to make it clear that rather than merely nonlinear grazing at low prey concentrations, a grazing function must increase with prey concentration faster than linearly at low prey concentrations. In mathematical terms, the second derivative of the grazing function with respect to phytoplankton should be positive as p goes to zero. We have included this analysis in section 2. This excludes the type of saturating nonlinearity that is present in a type II functional response. We have made this more clear in the abstract, in the results section (changing "non-linear" to "superlinear"). Discussion changed "there must be non-linearity in the interaction between the zooplankton and the phytoplankton at low phytoplankton concentrations" to "the grazing rate as a function of phytoplankton concentration must decrease faster than linearly" Changed "a nonlinear coupling between phytoplankton and zooplankton" to "the reduction in grazing rate".

*My major issue with Sec. 2 results is that the authors made the arbitrary assumption that the Holling type II is equivalent to a linear dependence on phytoplankton concentration, which is only true in the approximation of P->0. (Incidentally, please note that it is not always clear which parameters the authors used in Sec. 2, so I would suggest to improve this description). Fig. 2 thus presents a comparison between the Holling type I and Holling type III formulations, without considering Holling type II.*

Fig 2 presents a comparison between the Holling type I, II, and III formulations. The low phytoplankton approximation of the functional forms used in section 2 is not used in Fig 2. The full functional forms are used over a wide range of phytoplankton concentrations. It is evident from this figure that the sublinear change in grazing rate at moderate phytoplankton concentrations that is present in both the Holling type II and III formulations does not have the same effect as the faster than linear change in grazing rate at low concentrations. As shown in this figure, the simplifications in section 2 allow us to expose the dependence of grazing on the mixed layer depth at low phytoplankton concentrations more clearly without missing any important dependence of grazing

rates on the mixed layer depth at large phytoplankton concentrations. The analysis in section 2 is independent of any parameters.

*The resulting tuned value of parameter p0 for the Holling type II formulation applied in Sec. 3 is rather large, which forces this grazing rate to be in the linear part of the curve (Fig. 1) for the lowest modelled values of phytoplankton. I am therefore not much confident about the robustness of the results shown in Sec. 3, as I suspect some degree of overfitting led to the observed difference between the two formulations. This comment is further elaborated in point 4 below.*

The value of p0 used here (15 mg C/m$^3$) is within the range of the equivalent parameter used in previous work e.g. 20 $\mu$mol C/L Aumont et al 2015 Table 1b (PISCES), 20 mg C/m$^3$ Laufkotter et al 2015 Table 6 (PELAGOS). The Holling type II formulation has two linear parts, the part that increases linearly with phytoplankton concentration at low phytoplankton concentration and the part that is nearly constant with phytoplankton concentration at high phytoplankton concentration. The highest degree of nonlinearity is between those extremes but this nonlinearity is slower than linear and so does not reduce the grazing as the mixed layer deepens. With regards to the overfitting, we would like to point out that the analysis in section 2 is independent of the parameter values. We have approached this question from the perspective of model structure and elaborate on that approach in response to point 4.

*2. Another major issue I have with this manuscript is the apparent naivety in the biogeochemistry formulation and the related terminology. With all due respect, the sentences in line 65 and line 209 would have made me reject the manuscript immediately, since there is no carbon content in any dissolved inorganic nutrient and nitrogen pool cannot be described in units of carbon. I truly believe in the value of theoretical simplified models, and therefore I may interpret what the authors meant with these statements (this is why in mathematical modelling there is a tendency to use non-dimensional formulations). Nonetheless it is a major mistake that would confuse the non-modeller, and it is found throughout the manuscript. This actually invalidates any use of other*

*nutrient-related parameters taken from the literature (because they are usually in units of nitrogen or with multiple "currencies"), unless the authors state in advance what is the stoichiometric relationship between nutrients and carbon in their compartments and adjust the values accordingly. For instance, they refer to the nitrate half-saturation constant used by Moore et al. (2001), but in their appendix A this value is in mmol/m3. I would recommend the authors to present their formulations using the appropriate units for the generic nutrient, eventually adopting the use of constant stoichiometric ratios if needed.*

We thank the reviewer for pointing out the ways in which this model formulation was not clear. The currency of the model is indeed nutrients. We have now reported the values as they are given in the literature and all terms in the nutrient equation are in nitrogen units. We have added a clarification sentence that comparison to the observations is done in carbon units. "Phytoplankton biomass is compared to the observations in carbon units and conversions between nitrogen and carbon units are performed using a Redfield ratio of 16:106."

*3. The authors make the statement that loss terms other than grazing are of lower orders, but I argued in point 1 above that this is a not fully justified statement. I would thus recommend to explain more in detail their choices for the model formulation. For instance, does phytoplankton mortality parameterize respiration and viral lysis? This is a rather important concept, since this parameter has been considered freely tunable (see my point 3 below).*

The point about the importance of viruses is an important and was also mentioned by the other reviewer. We have now clarified that our model does not include viruses, but the effect of viruses is likely similar to the grazing functions that we have examined. We have added the following sentences to the section introducing the model. "The grazing function represents a density-dependent mortality process. Other mortality processes such as viruses may also be density-dependent and could be studied in a similar manner but are not included in this analysis (Weitz et al 2015, Mateus 2017). Some of the

conclusions may be applicable to other density-dependent mortality processes."

*I would also argue that the authors have not given much attention to the choice of the zooplankton closure formulation. Is there a specific reason for not using a linear mortality as indicated by Edwards and Yool (2000)? This is briefly mentioned at line 155 but without context it is difficult to appreciate its role. The disturbance-recovery hypothesis would also affect zooplankton mortality rates, which I understand was the authors' interpretation since they treated this parameter as freely tunable. However, if this parameter is re-tuned when the mathematical term is changed, then it is difficult to assess the outcome.*

We have chosen a quadratic mortality because it represents predation processes on zooplankton. This seems appropriate given the focusing of this article on density dependent processes. The results of Edwards and Yool 2000 do not provide convincing evidence that a linear mortality term is superior to a quadratic zooplankton mortality term. However, the reviewer is correct that dilution does affect zooplankton mortality in addition to zooplankton growth. This is discussed on lines 158-164. The focus of this manuscript is on the functional forms rather than the parameter values, as is explained in response to point 4.

*In addition, the authors do not always explain the use of certain mathematical notations. For instance, they use material derivatives in eq. (1) but make no assumptions as to why they are used and eventually turned into partial derivatives in eq. (2).*

We have corrected this error and now use material derivatives throughout.

*4. Another major point is linked to assuming the phytoplankton mortality constant (dp) dependent on the mathematical form of the grazing rate. If this loss term is meant to parameterise phytoplankton mortality other than grazing, should not this be treated as independent of zooplankton and hence belonging to the fixed set of parameters in Table 1? This is a first-order loss term, which is usually interpreted as basal (or biomass-related) metabolic losses. It may indeed also include viral lysis, although in*

*this case it should be density-dependent (see the excellent review in Mateus, 2017). I would recommend the authors to repeat their experiments assuming a constant value for this specific loss term and simply attributing it to basal respiration, which is the more plausible metabolic rate. It is not completely clear to me if the parameters calibrated in Table 1 have been used for the theoretical analysis presented in Sec. 2 and Fig. 2. Edwards and Yool (2000) gave clear indication that the role of parameters is often more important than the choice of the mathematical formulations, and it should be demonstrated that it is not the case here. In relation to Sec. 3, what would be very important, is to ensure that the overwintering phytoplankton biomass is the same in both the experiments, and that all parameters but the ones related to grazing terms are fixed. I do not agree much with the statement that this is an unknown (and thus freely-tunable) term. They can find a range of values in Lopez-Sandoval et al. (2014; between 9 and 22).*

We have set bounds on the parameter values that can result from the parameter fitting to ensure that the values are within the previously reported range of population-level loss rates. However, because this is an oversimplified model that encompasses a wide diversity in a single phytoplankton type, we cannot know the parameters exactly. The best we can do is fit the parameters to the available data. Our approach in this manuscript is to start with the model structure rather than the model parameters. If we try to fit a model to the observations and show that it cannot match the key features of the observations, then we must reject the model. If the model does fit, on the other hand, then we can examine the model with parameter values and assess if the model fits within a reasonable parameter range. The mathematical analysis in section 2 is independent of the parameter values and therefore shows that certain grazing functions cannot result in wintertime biomass accumulation under the other model assumptions given here regardless of the parameter values. However, the parameters have the same significance as in Table 1.

*At line 139 the authors state that mortality time scales are 10 times the scale of divi-*

*sion rates. This is an unclear and unsupported statement (division rate can be made equivalent to a carbon fixation, or production, rate using some algebra, but I think the authors should compare like with like). The paper cited above reports it to be about 10*

We have added additional support about the importance of grazing: "However the assumption is likely inappropriate for most blooms where grazing is a main source of mortality immediately prior to bloom formation (Calbet and Landry 2004, Irigoien et al 2005)."

*5. The dilution hypothesis is one of the possible explanations for winter accumulation of primary producer biomass, however this is based on the assumptions that light limitation is preclusive of any photon assimilation. Even if I did not expect the authors to be aware of the very recent papers on phytoplankton phenology cited above, I would have expected this work to give a more proper consideration to the role of light as a driver of phytoplankton growth. Not all the assumptions are justified, and particularly how the various processes have been averaged in the bulk mixed layer model. The authors added an implicit treatment of patchiness due to the entrainment of phytoplankton-depleted water from the deeper ocean, but ignored and did not discuss the inhomogeneity of growth due to the exponential decay of light in the mixed layer. This implies that the authors assume that the average phytoplankton growth is determined by the average light, thus assuming negligible that the Sverdrup himself suggested that growth may take place even with strong turbulence if the phytoplankton displays a positive phototaxis. This problem has been treated extensively in Paparella and Vichi (2020), both considering cases in which the biological rates are quicker than the physical scales and the opposite, which is the main founding assumption done by Sverdrup and also made by the authors. In both cases, it has*

In the float profiles used in this analysis we do not see a strong vertical dependence of chlorophyll in the mixed layer. Therefore, we assume that the biological variables are sufficiently well-mixed to justify the use of the zero-dimensional bulk mixed layer model and neglect 1-D effects. This is a simplification that yields relevant insights but which

necessarily does not capture all relevant aspects of the system.

*6. Finally, I'm afraid the authors gave a somewhat biased interpretation of the discussion about the Sverdrup model and instrumentally presented it to reinforce their argumentations. It is now widely accepted that the Sverdrup model is a conceptual mathematical formulation that helped to promote the quantitative study of the ocean microbiome, in an era when marine biogeochemistry was still a branch of descriptive biology. This is clearly illustrated in the cited paper by Fischer et al, and even more in the special issue of the ICES journal edited by Sathyendranath et al. (2015). Citing from the abstract, this set of papers celebrates an elegant and powerful hypothesis that has had long-lasting influence. Sverdrup clearly intended to describe the April-May period and not the winter accumulation, and therefore it should not be used as an alternative hypothesis for the winter case. It is also important to remark that none of the current biogeochemical models make predictions based on the Sverdrup hypothesis. As such, this is more to be seen as an emergent property, rather than an explanatory mechanism. I have thus been quite surprised to see that the authors made their own revisitation of the Sverdrup model in Sec. 2.1, attributing their interpretation to the original formulation. Sverdrup introduced just a first-order loss term and then, eventually, made a few considerations about the role of grazing, showing that the data available at that time were in agreement with his postulation. It is the authors' interpretation to attribute this constant loss term to grazing, and it should be made very clear. I would particularly recommend removing it from the title of sec. 2.1 because it would be rather misleading to those readers who are less familiar with the original paper.*

We have removed "Sverdrup" from the title of section 2.1 and replaced it with "critical depth hypothesis". We have also changed the text in this section to distinguish between Sverdrup 1953 and the critical depth hypothesis. This section does not claim that Sverdrup was wrong, but merely incomplete. In the spring when conditions change rapidly the critical depth hypothesis and Sverdrup 1953's conclusions are probably accurate. However, there are some limitations, as we state "While the critical depth

hypothesis has become the most widely accepted framework to interpret the onset of spring blooms–but there are growing objections (Behrenfeld 2010)–it is not very useful to make quantitative predictions. The criterion requires knowledge of the grazing rate at the end of winter before bloom onset, which is very difficult to measure."

*2 Specific comments L143 Concentration*

Thank you. We have corrected this typo.

*L152 (and L155-157) A decrease in MLD would likely enhance growth terms more than the grazing losses (see point 5 above). Since episodes of restratification are actually frequent in the ocean (e.g. Smith et al., 2016), this assumption may not hold even during fully mixed conditions.*

We selected float data that does not include lateral restratification (line 190) to avoid those effects. The statements on the cited lines refer to the specific model presented in this section and are not general statements about phytoplankton growth and physiology.

*L177 Please indicate which equation set is being used here*

We have added "equation" to the equation reference on line 180 to make it clear which equation is being used.

*L183 More details are needed to explain the modelling of the nutrient concentration and how this has been constrained according to observations. The use of a generic nutrient variable is sensible in Sec. 2, but here, because of the connection with the specific BGC-Argo measurements, it is important to clarify which nutrient is being used and how it has been assessed in terms of seasonal cycle. The floats do not measure nitrate, but other historical data can be used.*

The nutrient concentrations have not been constrained to observations. This is an idealized model and as subsequent sections explain we fit the model to phytoplankton accumulation rates only, with no assessment of fit to nutrients or zooplankton. We are

able to convey the key points about the dependence of grazing dynamics on the mixed layer depth without explicitly fitting the model to nutrients.

*Biology has been demonstrated to be patchier than physics in the North Atlantic (e.g. Mahadevan and Campbell, 2002). Please consider this in your discussion.*

We have changed "regions" in this sentence to "trajectories", which is more precise. We have also added citations to Mahadevan et al 2012 and Karimpour et al 2018 to the previous sentence. We have discussed some of the vertical velocity processes that cause patchy productivity in the North Atlantic "Finally, there is evidence that wintertime growth can be triggered by mixed layer instabilities that occasionally restratify the mixed layer during the winter and thus increase the light available for phytoplankton (Karimpour et al 2018). However this cannot be the unique explanation, because float observations presented in Mignot et al (2018) and reviewed here show many examples of wintertime accumulation where these mixed layer dynamics did not seem to apply."

*L192-193. There are not many figures in this manuscript. I would thus suggest the authors to add some more information on the floats that have been used, and possibly provide a table with their identification numbers. It would be valuable to see the individual timeseries, to appreciate the phenological variability prior to their standardization.*

These time series are available and we have added a reference to where they can be found. "These individual float trajectories are plotted in the appendix of (Mignot et al 2018)."

*L199,L202, and others. I would recommend the authors to be consistent in their terminology. There is a tendency to consider the term accumulation and growth as synonyms, however, there can be accumulation without growth.*

We have corrected this oversight.

*L205 I would need some more explanation here. According to the description, I was expecting a non-dimensional axis and not calendar days. If the model was not rescaled,*

*what are the t1 and t2 values for the model (and actually, having two different model formulations, which one was chosen?)*

We have clarified by writing in the previous sentence "where t1 is the calendar day of. . ."

*L217 I understand that this is more of a personal choice, but I think there is value in introducing the methodology at the beginning of the paper. The parameter fitting procedure is now interrupting the presentation of the results. Also, the actual description of the optimization procedure is shown later at lines 224-227. Many of my major issues are related to the possible influence of overfitting that may have biased the results, and I think the authors should make an effort to demonstrate that this is not the case. I am not much familiar with this methodology and therefore it would be useful to know if the parameters have been tested one-at-a-time and what kind of sensitivity analysis was performed.*

We have consolidated the presentation of the parameter fitting to a single paragraph and added additional details. The parameter fitting algorithm is well-suited for both linear and non-linear problems. The optimal solution is found iteratively within the whole parameter space rather than one parameter at a time. We performed a sensitivity analysis by changing the initial parameter combinations within the range of reasonable parameters.

*L231-232. Please indicate if eq. (12) was used to estimate this rate. I struggle to understand why the rate is positive but concentration keeps decreasing.*

Yes, equation 12 was used to estimate this rate, which is a biomass accumulation rate. The concentration is decreasing because the mixed layer is deepening. Although the biomass is increasing, the concentration is still decreasing because the dilution effect outpaces the accumulation effect.

*L273-276 My concerns on this statement are expressed in the major points above.*

*The degree of nonlinearity of the Holling type II formulation depends on the value of the parameter, which is tuned differently in the two experiments, as well as the basal loss rates. The response of zooplankton would largely depend on the extant biomass during winter, which is quite different in the two simulations.*

We have revised this sentence to read "We demonstrated that the grazing rate as a function of phytoplankton concentration must decrease faster than linearly at low phytoplankton concentrations in order to release the phytoplankton from grazing pressure."

*L294-295 Light dependence was made less influential by assumption (see point 6 above). Please rephrase.*

We have rephrased from "light availability" to "maximum insolation" to be more clear about what is meant here. The change in mixed layer depth is the main effect on light modification of growth.

---

## Referee Comment (RC3) · Anonymous Referee #3 · 23 Mar 2021

This study analyzes the importance of considering non-linear functional responses of grazing at low phytoplankton concentrations when modelling plankton dynamics. In particular, the authors point out that including these types of responses is key to reproduce the accumulation of phytoplankton biomass observed in winter in the North Atlantic. The manuscript is well written and the results and conclusions are interesting. However, I have some comments and questions that I think should be addressed in order to be published.

General comments:

1) If I understood correctly, in the study the phytoplankton specific growth rate decays exponentially with depth due to light absorption with an attenuation coefficient Kd. This would mean that the response of phytoplankton growth to light only depends on the surface irradiance, the Kd, and depth; i.e. depends on the light level at a particular depth. However, it seems that this dependency is modeled as a linear response. If this is the case, please consider that P-I curves have a non-linear form, expressed as a saturating response, or a curve with an optimum due to photoinhibition (see examples in Tian 2006). Although the response might be close to linear in winter due to low irradiance levels, non-linear responses might be important later in the year.

2) I could not find in the model how the effect of temperature on growth and grazing rates was introduced. The potential consequences of this effect were not considered in the discussion either. According to Rose and Caron (2007), low temperatures might im-pact more negatively microzooplankton grazing rates than phytoplankton growth rates (although see Chen et al. 2012), which can allow phytoplankton biomass accumula-tion in winter. Considering this, could a combination of temperature effect and linear grazing functional response allow a phytoplankton biomass accumulation in winter? Could this combination lead to similar results as those found when applying a grazing response that is non-linear at low phytoplankton concentrations?

3) Using dilution experiments, Liu et al. (2021) showed that "Holling III function best described the functional response of microzooplankton grazing" and highlighted the importance o this type of response at low phytoplankton concentrations. I think this paper or similar ones based on experimental observations support the results of the current study and should be mentioned in the discussion.

Specific comments:

About the title: maybe replace "An investigation of" with "Investigating" or "Analyzing." L18-24: I think at some point here, the Critical Turbulence Hypothesis (Huisman et al. 1999) could be also mentioned as it is a famous and important one.

L25: I would rather say that the Disturbance Recovery Hypothesis focuses on both phytoplankton growth and loss rates and how they are coupled or decoupled (i.e. on how their equilibrium is disrupted).

L34: What do you mean with loss at large scales? Please elaborate. Also, I think you could include a reference for this.

L36: Loss due to grazing also depends on temperature and probably on other environmental factors (see for instance Chen et al. 2012).

L40: It sounds like it is only possible to quantify this interaction through mathematical models. What about dilution experiments for example? It would be clearer if you say that it can be modeled through a mathematical relationship.

Fig. 2: What are the units of the axes? Also, I am a bit confused about what the contour colors represent. At the beginning of the figure caption, it says that colors represent grazing rates and in the next sentence, it seems that colors represent the rate of change in biomass. Additionally, in the case of Holling type III, for each phytoplankton concentration, rather than a decrease in the grazing rate with deeper mixed layer depths, there is first an increase and then a decrease (i.e. It seems that there is an optimal mixed layer depth for grazing rates at each phytoplankton biomass). Finally, I think the last sentences of the caption should be better written. Decreases and increases do not occur at a particular level but rather when moving along a particular axis (see for example "This occurs at low values of phytoplankton biomass and deep mixed layers" or "At high biomass there is also a decrease in grazing rate). At a particular combination of mixed layer depth and phytoplankton biomass can occur larger/est or lower/est grazing rates.

L 214: Reference for Kd = 0.05 m-1?

L 219-202: Maybe include a reference to Fig 3 saying between which days this peak is found. Fig. 3: There are too many lines in the gray grid. Select just a few for the x and

y axes. Why the grazing rate is not another panel? The axis labels are too small. In the units of the axis labels, erase the space before the exponent and separate mg and C. I'd add a vertical thicker line on day 1 to make it clear that the plots do not start on day 1. The thin black line from day 315 to day 5 is very difficult to see and can be confused with the thicker black line. Maybe use another color (blue?) and maybe make it dash.

L 238-239: Is it discussed whether the dp inferred by type III is more realistic? This could be supported with references.

L246: clarify which period is the end of winter by adding in parentheses which day/s of the year (or period in days of the year).

Fig. 4: Separate mg and C in the axis labels. Why for one of the curves there is a labeled dot for day 135 and in the other for 130? If there is not a clear justification, use the same day for better comparison.

L294-295: Why does light little influence on wintertime biomass accumulation? Does not an increase in light through the seasonal cycle increase phytoplankton specific growth rates and contribute to the decoupling with their grazers?

Technical corrections

L133: modify reference as "(Behrenfeld, 2010)" and maybe introduce it as "(see for example Behrenfeld, 2010)".

L136: Comma after "However".

L343: very "difficult" to quantify?

References

Chen, B., M. R. Landry, B. Huang, and H. Liu. 2012. Does warming enhance the effect of microzooplankton grazing on marine phytoplankton in the ocean? 57: 519-526.

Huisman, J., P. van Oostveen, and F. J. Weissing. 1999. Critical depth and critical

turbulence: two different mechanisms for the development of phytoplankton blooms. Limnol. Oceanogr. 44: 1781-1787.

Liu, K. and others 2021. What controls microzooplankton biomass and herbivory rate across marginal seas of China? 66: 61-75.

Rose, J. M., and D. A. Caron. 2007. Does low temperature constrain the growth rates of heterotrophic protists? Evidence and implications for algal blooms in cold waters. 52: 886-895.

Tian, R. C. 2006. Toward standard parameterizations in marine biological modeling. Ecol. Model. 193: 363-386.

---

## Author Comment (AC3) · 21 May 2021

This study analyzes the importance of considering non-linear functional responses of grazing at low phytoplankton concentrations when modelling plankton dynamics. In particular, the authors point out that including these types of responses is key to re-produce the accumulation of phytoplankton biomass observed in winter in the North Atlantic. The manuscript is well written and the results and conclusions are interesting. However, I have some comments and questions that I think should be addressed in order to be published.

REPLY: We thank the reviewer for their positive assessment of our work.

General comments: 1) If I understood correctly, in the study the phytoplankton specific growth rate decays exponentially with depth due to light absorption with an attenuation coefficient Kd. This would mean that the response of phytoplankton growth to light only depends on the surface irradiance, the Kd, and depth; i.e. depends on the light level at a particular depth. However, it seems that this dependency is modeled as a linear response. If this is the case, please consider that P-I curves have a non-linear form, expressed as a saturating response, or a curve with an optimum due to photoinhibition (see examples in Tian 2006). Although the response might be close to linear in winter due to low irradiance levels, non-linear responses might be important later in the year.

REPLY: It is true that a saturating irradiance model might affect the full annual cycle, especially in the spring-summer. However, the reviewer is correct that we assumed that the response is close to linear during the winter due to low irradiance levels. Since our focus is indeed on the winter period, we have chosen to use the linear function to reduce the number of parameters in the model. Given this focus on the winter period when light levels are low, non-linear dependence of growth on light does not affect our core message. We will discuss this assumption more explicitly when presenting the full annual cycle in the revised manuscript.

2) I could not find in the model how the effect of temperature on growth and grazing rates was introduced. The potential consequences of this effect were not considered in the discussion either. According to Rose and Caron (2007), low temperatures might impact more negatively microzooplankton grazing rates than phytoplankton growth rates (although see Chen et al. 2012), which can allow phytoplankton biomass accumulation in winter. Considering this, could a combination of temperature effect and linear grazing functional response allow a phytoplankton biomass accumulation in winter? Could this combination lead to similar results as those found when applying a grazing response that is non-linear at low phytoplankton concentrations?

REPLY: The reviewer brings up an interesting point regarding correlation between phytoplankton concentrations and temperature. We did not include temperature explicitly in the model, but we do include a section in the discussion about possible effects of other time-dependent terms. If we assume that temperature is approximately proportional to the mixed layer depth and if indeed it only affects grazing rates but not phytoplankton growth rates then we will see that there is an apparent release from grazing as the mixed deepens even when using a linear model. We have chosen to use a simplified model that does not include all potentially relevant factors to make progress towards improved understanding. This comment reveals that additional insights are likely possible using the framework that we have outlined in the manuscript. We focused on grazing because of the support in the literature for this as a potential mechanism for wintertime biomass accumulation (Behrenfeld 2010). There is inconsistent support for temperature dependence of grazing rates that would trigger an accumulation of phytoplankton in winter (Rose and Caron 2007, Lopez-Urrutia 2008, Chen et al 2012). We discuss alternate mechanisms for wintertime biomass accumulation in the paragraph beginning on line 291 and will add additional discussion of temperature effects to that paragraph.

3) Using dilution experiments, Liu et al. (2021) showed that "Holling III function best described the functional response of microzooplankton grazing" and highlighted the importance o this type of response at low phytoplankton concentrations. I think this paper or similar ones based on experimental observations support the results of the current study and should be mentioned in the discussion.

REPLY: Thank you very much for pointing out this recent reference which provides additional experimental constraints on grazing functional responses. We will include a citation to this study, which supports our findings, in the discussion.

Specific comments: About the title: maybe replace "An investigation of" with "Investigating" or "Analyzing."

REPLY: Based on this comment we will simplify the title to "Grazing behavior and winter phytoplankton accumulation"

L18-24: I think at some point here, the Critical Turbulence Hypothesis (Huisman et al. 1999) could be also mentioned as it is a famous and important one.

REPLY: This is a good suggestion. In the revised introduction we will include the critical turbulence hypothesis. When doing so, we will also include references to the recent work on positive phototaxis and the relative importance of biological and physical timescales as mentioned by reviewer #2 (point 5).

L25: I would rather say that the Disturbance Recovery Hypothesis focuses on both phytoplankton growth and loss rates and how they are coupled or decoupled (i.e. on how their equilibrium is disrupted).

REPLY: We have changed this sentence from "An alternative hypothesis proposed by Behrenfeld 2010 focuses on changes in loss rates rather than growth rates." to "An alternative hypothesis proposed by Behrenfeld (2010) focuses on changes in both loss rates and growth rates."

L34: What do you mean with loss at large scales? Please elaborate. Also, I think you could include a reference for this.

REPLY: By "at large scales" here we mean that in situ observations are needed for measuring zooplankton distributions and grazing rates. This is in contrast to the widespread autonomous measurements of nutrients, light, and chlorophyll concentration. This wording is confusing and we will change it to "for a whole population".

L36: Loss due to grazing also depends on temperature and probably on other environmental factors (see for instance Chen et al. 2012).

REPLY: We will revise this sentence to clarify that we did not mean that loss due to grazing depends exclusively on phytoplankton and zooplankton concentration, but we focus on that particular dependence because its significance has not been fully appreciated.

L40: It sounds like it is only possible to quantify this interaction through mathematical models. What about dilution experiments for example? It would be clearer if you say that it can be modeled through a mathematical relationship.

REPLY: As the reviewer suggests, we will change "quantified" to "modeled"

Fig. 2: What are the units of the axes? Also, I am a bit confused about what the contour colors represent. At the beginning of the figure caption, it says that colors represent grazing rates and in the next sentence, it seems that colors represent the rate of change in biomass. Additionally, in the case of Holling type III, for each phytoplankton concentration, rather than a decrease in the grazing rate with deeper mixed layer depths, there is first an increase and then a decrease (i.e. It seems that there is an optimal mixed layer depth for grazing rates at each phytoplankton biomass). Finally, I think the last sentences of the caption should be better written. Decreases and increases do not occur at a particular level but rather when moving along a particular axis (see for example "This occurs at low values of phytoplankton biomass and deep mixed layers" or "At high biomass there is also a decrease in grazing rate"). At a particular combination of mixed layer depth and phytoplankton biomass can occur larger/est or lower/est grazing rates.

REPLY: Both this and the other reviewers have highlighted ways in which figure 2 is unclear. Rather than just updating the caption, in the revision we will improve the clarity of this figure to better make the main point that the dependence of the grazing rate on the mixed layer depth (and phytoplankton biomass) differs for the various grazing functional responses. To answer the specific question of the reviewer, the axis units are meters for mixed layer depth and mg C per meter squared for biomass.

L 214: Reference for Kd = 0.05 m-1?

REPLY: The reference for this is Organelli et al 2017, which presents a global map of

attenuation coefficients from Argo floats. This value is also computed for the set of floats used in this manuscript in Mignot et al 2018. These references will be given in the revised manuscript.

L 219-202: Maybe include a reference to Fig 3 saying between which days this peak is found.

REPLY: We have added a reference to Fig 3.

Fig. 3: There are too many lines in the gray grid. Select just a few for the x and y axes. Why the grazing rate is not another panel? The axis labels are too small. In the units of the axis labels, erase the space before the exponent and separate mg and C. I'd add a vertical thicker line on day 1 to make it clear that the plots do not start on day 1. The thin black line from day 315 to day 5 is very difficult to see and can be confused with the thicker black line. Maybe use another color (blue?) and maybe make it dash.

REPLY: In the revised version we will improve the clarity of this figure

L 238-239: Is it discussed whether the dp inferred by type III is more realistic? This could be supported with references.

REPLY: These values are problematic to compare for different grazing formulations because the linear mortality includes different processes if the grazing is parameterized differently. One process that is included in the linear mortality is phytoplankton respiration. Phytoplankton respiration rates from in situ observations and incubation experiments fall within the range of these linear mortality rate estimations from the parameter fitting (Lopez-Sandoval et al 2014, Briggs et al 2018). We will add this discussion and these references to the manuscript.

Briggs, N., GuÃřmundsson, K., Cetinić, I., D'Asaro, E., Rehm, E., Lee, C., & Perry, M. J. (2018). A multi-method autonomous assessment of primary productivity and export efficiency in the springtime North Atlantic. Biogeosciences, 15(14), 4515-4532.

L246: clarify which period is the end of winter by adding in parentheses which day/s of

the year (or period in days of the year).

REPLY: We have specified the days: 320-365 and continuing to 1-75.

Fig. 4: Separate mg and C in the axis labels. Why for one of the curves there is a labeled dot for day 135 and in the other for 130? If there is not a clear justification, use the same day for better comparison.

REPLY: Different days were labeled to reduce the cluttering on the figure but we will instead use the same day for comparison.

L294-295: Why does light little influence on wintertime biomass accumulation? Does not an increase in light through the seasonal cycle increase phytoplankton specific growth rates and contribute to the decoupling with their grazers?

REPLY: It is true that light has an influence on the annual cycle, but the phenomenon of wintertime biomass accumulation does not arise from the annual cycle in light. The light is decreasing or low during the winter. Moreover, the changes in light are the same with the two functional responses.

Technical corrections L133: modify reference as "(Behrenfeld, 2010)" and maybe introduce it as "(see for example Behrenfeld, 2010)".

REPLY: We have made the suggested modification.

L136: Comma after "However".

REPLY: We have made this correction

L343: very "difficult" to quantify?

REPLY: Yes. We have added this missing word

---

## Author Comment (AC4) · 21 May 2021

We again thank the reviewer for their thoughtful comments, which, as the reviewer states, will help ensure that the results are interpreted within the proper context and with sufficient support from the literature. After reflecting on the comments from all three reviewers, we have some additional answers on the reviewer's main comments in addition to the ones we previously posted.

REPLY to 1. We have added additional support to the introduction for the focus on grazing. For example, we have added the sentence "Phytoplankton are thought to be

tightly controlled by grazing and loss processes within the microbial food web (Landry and Calbet 2004, Calbet and Landry 2004, Strom et al 2007, Evans et al 2012, Prowe et al 2012, Liu et al 2021). Any accumulation depends on the imbalance between growth and loss processes, which have been shown to be tightly correlated (Behrenfeld and Boss 2017)." to the fourth paragraph of the introduction. Given the limited availability of observations of grazing, particularly in regions with deep mixed layers in the wintertime, we rely on both models and observations to make the case for the relevance of top-down control in the wintertime. Following the reviewer comment we worked on making our terminology more precise. We have edited the manuscript text throughout to make it clear that the grazing function must increase with prey concentration faster than linearly at low prey concentrations. This is distinct from the grazing function being generally nonlinear and is an important clarification. An additional key point is that the prey concentrations must be low enough that the grazing rate is in the region of the grazing response function where grazing increases superlinearly with prey concentration. If the wintertime conditions were still in the saturated part of either the type II or type III grazing rate function it would not give wintertime accumulation. Figure 2 is meant to summarize the key points and extend the theoretical analysis that was performed near $p = 0$ to the full functions. However, the reviewer comments made us realize that this figure needs to be revised. In our revised manuscript, we will replace the current figure 2 with a comparison of the response of growth and grazing to mixed layer depth. This figure will support our main point that the dependence of the grazing rate on the mixed layer depth (and phytoplankton biomass) is distinct between the grazing functional responses. Although the type II and type III models have different shapes, they have the same number of parameters. This allows us to make a more robust comparison between the two functions when we perform the parameter fitting without biasing the results to one or the other function.

REPLY to 3. The reviewer proposes potential modifications to the model or additional processes that could be included. Viral lysis is indeed an interesting and important process. Our model does not explicitly represent viral lysis but from a mathematical

standpoint viral lysis is often represented using grazing functions similar to the ones we have examined. While inclusion of viral lysis in a similar framework is a potentially compelling follow-up to this study, we do not see an effect of viral lysis reported in the literature that invalidates our key result about the grazing formulation. As for the specifics of the parameter values, the focus of this manuscript is on the functional forms rather than the parameter values. The theoretical analysis is independent of parameter values.

REPLY to 4. In this study we use a simplified model that encompasses a wide range of phytoplankton species in a single phytoplankton type. Therefore, we cannot easily infer the parameters from culture studies. The fitted values for the phytoplankton mortality are within the range of values in Lopez-Sandoval et al 2014, but they vary by an order of magnitude for each functional form. Allowing for this range of variability avoids unnecessarily constraining the population dynamics into an ill-suited model parameter space. Instead, by beginning with the model structure and fitting the parameters, we are able to compare the grazing processes in a way that allows us to best evaluate the influence of model structure. We would like to reiterate a statement made in the previous response because it is a key point about our approach. If we try to fit a model to the observations and show that it cannot match key features of the observations for any parameter values, then we must reject that model. If the model can be tuned to fit the observations, then we can examine the model and parameter values together to assess if the model fits within a reasonable parameter range. In the revised draft, we will more transparently work through this method to place our results in their appropriate context.

REPLY to 5. There are some assumptions of the model, particularly the bulk mixed layer formulation, which may fail at times when mixing is weak or intermittent. In the revised introduction we will include a thorough treatment of phytoplankton growth including the Critical Turbulence Hypothesis (Huisman et al. 1999), positive phototaxis, and the relative importance of biological and physical timescales. This is relevant to

address because the essence of the theoretical contribution is that the relative dependence of growth and grazing on the mixed layer depth determines the response to dilution. The studies cited by the reviewer support the relatively weak dependence of phytoplankton growth compared to grazing on the mixed layer depth. More generally, we wish to reiterate that the contribution of our manuscript is to show that blooms triggered by deepening of a mixed layer and the associated dilution of phytoplankton can develop only for specific grazing functions. This point is treated more extensively in our general comment.

---

## Author Comment (AC5) · 21 May 2021

We appreciate the careful reading and thoughtful comments by all three reviewers. These comments have helped us to improve the clarity of the presentation and to refine the discussion of the main contribution of this manuscript. Following the reviewer comments, we have worked on making our terminology more precise and clarifying the presentation of the main results. The essence of the contribution of our manuscript is that growth triggered by a reduction in grazing pressure through dilution of plankton populations in a deepening mixed layer can develop only for specific grazing functions. This has not been pointed out in the literature, despite much attention to the dilution

effect. In the manuscript we use simplified models to illustrate this key point. Based on their comments, all reviewers appear to be convinced by our analysis and conclusions. The comments primarily focus on a wide variety of additional factors that could and do contribute to the annual cycle of phytoplankton growth and wintertime phytoplankton biomass accumulation rates. These comments are well-taken; we appreciate the complexity of the system and any model of a biological system is incomplete. Our approach is not to propose a complete set of processes that should be considered but instead to advance understanding of a limited set of processes. We base our methods on the perspective that to be useful models should be as simple as possible but not simpler. We recognize that additional processes can increase phytoplankton growth or reduce grazing in a less direct way than the simple functional forms. These will not invalidate the basic results that grazing or other sources of mortality must decrease more rapidly than linear for small prey concentrations. In the detailed responses we explain that all the additional processes highlighted by the reviewers do not seem to impact our key conclusions. The manuscript, however, is neither addressing nor rejecting hypotheses of wintertime accumulation due to alternative processes. There are a number of models that do include a wide range of biological processes. These models are implemented in global and regional models and used to make predictions about ecosystem responses to global change. Many of these models are moving towards non-linear grazing functions. We demonstrate that that choice is quite important for wintertime biomass accumulation, a point that has not been made explicit in the literature.

We have provided detailed responses to each reviewer in separate comments.

---

## Author Response (AR1)

Comment #1: [Review]

The manuscript deals with the mathematical description of grazing behaviors which may explain the (weak) increase in phytoplankton biomass observed in winter in the North Atlantic Ocean. The positive growth of phytoplankton during winter, light limiting conditions was previously explained by invoking a decrease in grazing pressure. More specifically, it was suggested that, due to the dilution of the water column caused be the deepening of the mixed layer, the encounter rate between phytoplankton and their grazers decreases, and this allows a weak positive growth despite light limitation. The authors highlight that this behavior can be modelled assuming a non-linear (in this case quadratic) relationship between grazing and prey biomass, a functional response known as Holling type III function. The main conclusion of the work is that assuming this latter functional response a simple biogeochemical model is able to simulate both the increase in biomass in winter and the subsequent spring bloom, while other widely used grazing functional responses underestimate winter phytoplankton biomass. The Holling type III model is not new and is, as also stated by the authors, already a quite popular choice within the marine modelling community. However, this paper provides a remarkably clear explanation on why this formulation should be preferred over the others, also highlighting its ecological significance. The manuscript is clearly written, model's assumptions are very well explained and the results are clear and logic. I recommend publishing this paper after minor revisions.

**REPLY: We thank the reviewer for their thoughtful comments. We agree with the reviewer about this manuscript's main contribution to the literature and appreciate the suggestions to improve it. We outline the changes we have made based on these comments below.**

I appreciate that this paper focuses on zooplankton. However, other phytoplankton loss terms can be equally important in the formation and progression of a blooms (including relatively high winter biomass). For example, viral infection, possibly one of the main causes of mortality after grazing, could also be described by a non-linear, density dependent functional response (e.g. Mateus, 2017, FMS), and have a reduced effect on phytoplankton at low biomass concentration.

**REPLY: As long as the mortality has a nonlinear dependence on P, our conclusions remain unchanged. It was indeed an oversight to neglect viruses in the discussion as they clearly contribute to phytoplankton mortality. Their inclusion would further strengthen our conclusions. We have added the following sentence in the presentation of the model "The grazing function represents a density-dependent mortality process. Other mortality processes such as viral lysis are also believed to be density-dependent and could be studied within the same framework (Weitz et al 2015, Mateus 2017). While we retrict the analysis to zooplankton grazing, our qualitative conclusions are likely to apply to other density-dependent mortality processes."**

The presented model follows the classic dichotomy between autotrophic and heterotrophic organisms. However, we know that most phytoplankton exhibits mixotrophic metabolism, perhaps with the important exception of diatoms (see e.g. Flynn et al 2012 JPR, Gonçalves

Leles et al., 2018 JPR and 2020 Progress in oceanography). I am wondering if mixotrophy could be involved in the increase of biomass observed under light limiting conditions.

**REPLY: We agree that mixotrophy could be involved in the increase of biomass during the wintertime. Further modeling and observational studies would be needed to evaluate the role of heterotrophy and mixotrophy in the wintertime North Atlantic. We have now included this point in the discussion by adding the following sentence on line 322: "Functional diversity beyond that included in this model is also likely important. For example, mixotrophic metabolisms may contribute to phytoplankton accumulation in light-limited conditions (Barton et al. 2013, Flynn et al. 2013, Gonçalves Leles et al. 2020)."**

Figure1 is a bit misleading. I think that the different functional responses (II and III) need to be run with the same parameters in order to clearly assess differences and similarities.

**REPLY: As it is currently presented Figure 1 aids with interpretation of Figures 3 and 4, which use the optimal parameters for each functional type. We need to use different parameters for each model otherwise the model for which the parameters are optimized will by definition perform better.**

Line 88: the term $1/k_dH(t)$ is "the average light over the mixed layer.." but light is not explicitly modelled, right? Is the specific growth rate which is averaged over the ML, assuming an exponential decay

**REPLY: It is correct that light is not explicitly modeled. We have revised this sentence for clarity to read "The term $\mu\_0/K_dH(t)$ is the average growth rate over the mixed layer, which is computed as the integral of the light-dependent growth over the mixed layer depth divided by the mixed layer depth."**

Line 110: the formulation (type III) "is quadratic in p for low p". Why only for low p? I see that the focus here is on low winter biomass values but the formulation is quadratic for any p.

**REPLY: We have modified this sentence for clarity. It now reads "is quadratic in p for low p and asymptotes to a constant rate for high p."**

Comment #2

General comments
This manuscript aims at explaining the phytoplankton growth rate observed during winter in the ocean mixed layer, when conditions are usually not conducive to biomass accumulation. The work is stimulated by the recent availability of data offered by the BGC-Argo platforms, which are providing unprecedented information on the open ocean microbiome. Positive winter growth rates have been recently identified using Argo floats under sea ice, both in the Arctic and in the Antarctic (Randelhoff et al., 2020; Hague and Vichi, 2020). This is an indication that, even in

extreme conditions of light limitations (and likely with minimal predation pressure), phytoplankton have developed the capacity to grow and accumulate biomass. It is therefore (nowadays) less surprising that this may occur in the winter mixed layer of the North Atlantic Ocean. It is clearly a balance between growth and loss terms, and we are only now - thanks to the new autonomous technologies - starting to understand some of the details of the phytoplanktonic component. This work is however timely and of relevance for the discussion, but I have some issues with the current manuscript, some of them that I consider quite relevant in order to gain confidence in the presented results. In its current form, I would recommend a thorough revision of the introduction, assumptions and the related methodologies, to ensure that the results are of wider applicability.

**REPLY: We thank the reviewer for their thorough and thoughtful comments. This manuscript focuses on the mechanisms underlying the wintertime biomass accumulation, which has previously been reported from autonomous observations. We have added the recent references to the introduction. We have made revisions and clarifications where appropriate, as outlined below.**

1. My first major comment is related to the focus on the grazing formulation in the context of winter biomass accumulation. To emphasize the importance of grazing in tilting the balance of winter phytoplankton development, the authors state that phytoplankton loss is due primarily to grazing and cellular lysis, and secondarily to respiration and natural mortality. Notwithstanding the quite ambiguous definition of "natural mortality", which should very likely include cellular lysis induced by viruses, they attribute this sentence to Landry and Calbet (2004). I would argue that the cited paper, as well as the twin publication Calbet and Landry (2004), specifically meant to highlight the role of microzooplankton grazing in the microbial food chain, through the analysis of dilution experiments. It is not a generic study on the grazing relationship between phytoplankton and their predators, neither specifically targeting the winter period, which is the focus of this manuscript. I would suggest the authors present their case with more support from the existing literature.

**We have added additional support to the introduction to justify the focus on grazing. For example, we have added the sentence "Phytoplankton are thought to be tightly controlled by grazing and loss processes (Landry and Calbet 2004, Calbet and Landry 2004, Strom et al 2007, Evans et al 2012, Prowe et al 2012, Liu et al 2021). Any accumulation depends on the imbalance between growth and loss processes (Behrenfeld and Boss 2017)." to the fourth paragraph of the introduction. Given the limited availability of observations of grazing, particularly in regions with deep mixed layers in the wintertime, we rely on both models and observations to make the case for the relevance of top-down control in the wintertime. We have added additional citations to observational programs that observed that top down control is important, although, given observational limitations at high latitudes that are addressed in the introduction, these are not wintertime observations. We have also included a citation to modeling work that reveals the importance of top down control.**

I would also recommend the authors to be more rigorous in their use of the concept of linear and nonlinear grazing terms, because I have found their terminology, as well as its use in the presentation of the results, confusing at times. The grazing term of the dynamics is always nonlinear (it is the product of the two state variables P and Z), while the formulation of the grazing rate can be linear. Apart from some theoretical works in the 80's and early 90's (which can be found in the cited work by Gentleman et al, 2003), there is not one single "state-of-the-art" biogeochemical model that uses a linear formulation for the specific grazing rate (at least to my knowledge. I checked PISCES, BFM, BioEBUS, MEDUSA and a few others and they implement various combinations of hyperbolic functions or Ivlev's. I'm afraid the current wording may mislead the reader to assume that the majority of models uses a linear formulation, and this is why they may fail to show the winter accumulation. Throughout the manuscript I started to realize that maybe the authors intended that the hyperbolic grazing rate function is linear at "low prey concentrations". I concede that this is true for biomass values close to 0, as can be seen with a Taylor series expansion of the term around 0. For any value larger than 0, it however depends on the choice of the grazing saturation factor.

**REPLY: The reviewer has made the valuable comment that the wording about the types of nonlinearity that can promote wintertime biomass accumulation should be made more precise. We have edited the manuscript text to make it clear that rather than it is not sufficient for the grazing to be any nonlinear function at low prey concentrations, the function must increase faster than linearly with prey concentration at low concentrations. This is an important clarification. An additional key point is that the prey concentrations must be low enough that the grazing rate is in the region of the grazing response function where grazing increases superlinearly with prey concentration. If the wintertime conditions were still in the saturated part of either the type II or type III grazing rate function, grazing would not give wintertime accumulation.  In mathematical terms, the second derivative of the grazing function with respect to phytoplankton should be positive as p goes to zero. This excludes the type of saturating nonlinearity that is present in a type II functional response. We have included these important points in section 2. We have also emphasized this aspect in the abstract and in the results section. We changed "non-linear" to "superlinear";"there must be non-linearity in the interaction between the zooplankton and the phytoplankton at low phytoplankton concentrations" to "the grazing rate as a function of phytoplankton concentration must decrease faster than linearly";**
**"a nonlinear coupling between phytoplankton and zooplankton" to "the reduction in grazing rate".**

My major issue with Sec. 2 results is that the authors made the arbitrary assumption that the Holling type II is equivalent to a linear dependence on phytoplankton concentration, which is only true in the approximation of P->0. (Incidentally, please note that it is not always clear which parameters the authors used in Sec. 2, so I would suggest to improve this description). Fig. 2 thus presents a comparison between the Holling type I and Holling type III formulations, without considering Holling type II.

**REPLY: Fig 2 presents a comparison between the Holling type I, II, and III formulations. Figure 2 is meant to summarize the key points and extend the theoretical analysis, which was performed near p = 0 to the full functions. However, the reviewer comments made us realize that this figure was not driving the main point home. In our revised manuscript, we have replaced the previous figure 2 with a comparison of the response of growth and grazing to mixed layer depth, which better aligns with our theoretical analysis in section 2. This figure supports our main point that the dependence of the grazing rate on the mixed layer depth (and phytoplankton biomass) is different between the grazing functional responses.**

**We prefer to compare the type II and type III grazing functions, because despite having different functional forms, they have the same number of parameters. This allows us to make a more robust comparison between the two functions when we perform the parameter fitting without biasing the results to one or the other function.**

The resulting tuned value of parameter p0 for the Holling type II formulation applied in Sec. 3 is rather large, which forces this grazing rate to be in the linear part of the curve (Fig. 1) for the lowest modelled values of phytoplankton. I am therefore not much confident about the robustness of the results shown in Sec. 3, as I suspect some degree of overfitting led to the observed difference between the two formulations. This comment is further elaborated in point 4 below.

**REPLY: The value of p0 used here (15 mg C/m^3) is within the range of the equivalent parameter used in previous work, e.g., 20 micromol C/L (Aumont et al 2015, Table 1b;PISCES), 20 mg C/m^3 (Laufkotter et al 2015, Table 6; PELAGOS). Figures 1 and 2 show that the type II and type III functional responses reach saturation at similar phytoplankton concentrations with the fitted values. With regards to overfitting, we would like to point out that the analysis in section 2 is independent of the parameter values. We have approached this question from the perspective of model structure and elaborate on that approach in response to point 4.**

2. Another major issue I have with this manuscript is the apparent naivety in the biogeochemistry formulation and the related terminology. With all due respect, the sentences in line 65 and line 209 would have made me reject the manuscript immediately, since there is no carbon content in any dissolved inorganic nutrient and nitrogen pool cannot be described in units of carbon. I truly believe in the value of theoretical simplified models, and therefore I may interpret what the authors meant with these statements (this is why in mathematical modelling there is a tendency to use non-dimensional formulations). Nonetheless it is a major mistake that would confuse the non-modeller, and it is found throughout the manuscript. This actually invalidates any use of other nutrient-related parameters taken from the literature (because they are usually in units of nitrogen or with multiple "currencies"), unless the authors state in advance what is the stoichiometric relationship between nutrients and carbon in their compartments and adjust the values accordingly. For instance, they refer to the nitrate half-saturation constant used by Moore et al. (2001), but in their appendix A this value is in mmol/m3. I would recommend the

authors to present their formulations using the appropriate units for the generic nutrient, eventually adopting the use of constant stoichiometric ratios if needed.

**REPLY: We thank the reviewer for pointing out the ways in which the model formulation was not accurate. The currency of the model is indeed nutrients. We have now reported the values as they are given in the literature and all terms in the nutrient equation are in nitrogen units. We have added a clarification sentence that comparison to the observations is done in carbon units. "Phytoplankton biomass is compared to the observations in carbon units and conversions between nitrogen and carbon units are performed using a Redfield ratio of 16:106."**

3. The authors make the statement that loss terms other than grazing are of lower orders, but I argued in point 1 above that this is a not fully justified statement. I would thus recommend to explain more in detail their choices for the model formulation. For instance, does phytoplankton mortality parameterize respiration and viral lysis? This is a rather important concept, since this parameter has been considered freely tunable (see my point 3 below).

**REPLY: The reviewer proposes potential modifications to the model and additional processes that could be included. Viral lysis is indeed an interesting and important process. Our model does not explicitly represent viral lysis but from a mathematical standpoint viral lysis is often represented using grazing functions similar to the ones we have examined. While inclusion of viral lysis in a similar framework is a potentially compelling follow-up to this study, addition of viral lysis as represented in the published literature would further support our key result about the grazing formulation. We have now clarified that our model does not include viral lysis, but its qualitative effect is likely similar to that of the grazing functions that we have examined. We have added the following sentences to the section introducing the model. "The grazing function represents a density-dependent mortality process. Other mortality processes such as viral lyasis are also believed to be density-dependent and could be studied within the same framework in (Weitz et al 2015, Mateus 2017). While we restrict the analysis to zooplankton grazing, our qualitative conclusions are likely to apply to other density-dependent mortality processes."**

I would also argue that the authors have not given much attention to the choice of the zooplankton closure formulation. Is there a specific reason for not using a linear mortality as indicated by Edwards and Yool (2000)? This is briefly mentioned at line 155 but without context it is difficult to appreciate its role. The disturbance-recovery hypothesis would also affect zooplankton mortality rates, which I understand was the authors' interpretation since they treated this parameter as freely tunable. However, if this parameter is re-tuned when the mathematical term is changed, then it is difficult to assess the outcome.

**REPLY: The focus of this manuscript is on the functional forms rather than the parameter values. The key results are qualitative and independent of the specific parameter values. We have chosen a quadratic mortality because it represents predation processes on**

**zooplankton. This seems appropriate given the focus of this article on concentration dependent processes. The results of Edwards and Yool 2000 do not provide convincing evidence that a linear mortality term is superior to a quadratic zooplankton mortality term. However, the reviewer is correct that dilution does affect zooplankton mortality in addition to zooplankton growth. This is discussed in the last paragraph of section 2.2.**

In addition, the authors do not always explain the use of certain mathematical notations. For instance, they use material derivatives in eq. (1) but make no assumptions as to why they are used and eventually turned into partial derivatives in eq. (2).

**REPLY: We have corrected this error and now use material derivatives throughout.**

4. Another major point is linked to assuming the phytoplankton mortality constant (dp) dependent on the mathematical form of the grazing rate. If this loss term is meant to parameterise phytoplankton mortality other than grazing, should not this be treated as independent of zooplankton and hence belonging to the fixed set of parameters in Table 1? This is a first-order loss term, which is usually interpreted as basal (or biomass-related) metabolic losses. It may indeed also include viral lysis, although in this case it should be density-dependent (see the excellent review in Mateus, 2017). I would recommend the authors to repeat their experiments assuming a constant value for this specific loss term and simply attributing it to basal respiration, which is the more plausible metabolic rate.
It is not completely clear to me if the parameters calibrated in Table 1 have been used for the theoretical analysis presented in Sec. 2 and Fig. 2. Edwards and Yool (2000) gave clear indication that the role of parameters is often more important than the choice of the mathematical formulations, and it should be demonstrated that it is not the case here. In relation to Sec. 3, what would be very important, is to ensure that the overwintering phytoplankton biomass is the same in both the experiments, and that all parameters but the ones related to grazing terms are fixed. I do not agree much with the statement that this is an unknown (and thus freely-tunable) term. They can find a range of values in Lopez-Sandoval et al. (2014; between 9 and 22).

**REPLY: In this study we use a simplified model that encompasses a wide range of phytoplankton species in a single phytoplankton type. Therefore, we cannot easily infer the parameters from culture studies. The fitted values for the phytoplankton mortality are within the range of values in Lopez-Sandoval et al 2014, but they vary by an order of magnitude for the two different functional forms. By fitting the parameters independently for the two functional forms, we are able to assess whether the deficiencies in the model stem from the functional form of the grazing function rather than the specific parameter values. If the model cannot reproduce key features of the observations for any parameter values, then the functional form of the model must be rejected. If the model can be tuned to fit the observations, then we can examine the model and parameter values together to assess if the model fits within a reasonable parameter range.  In the revised draft, we tried to explain more clearly this method to place our results in their appropriate context by writing "The focus of this manuscript is on the functional formulation of the model.If**

the model cannot reproduce the key features of the observations for any values of the parameters, then the model must be rejected. If we can find parameter values for which the model reproduces key features of the observations, we then assess if those values are consistent with observational estimates." in the first paragraph of the "model parameters" section.

**We have set bounds on the parameter values that can result from the parameter fitting exercise to ensure that the values are within those reported in the literature for population-level loss rates. We explore a wide range of parameter values because we use an oversimplified model that encompasses a wide diversity in a single phytoplankton type and we cannot use values for any specific species. Alternatively we find the parameters that best reproduce the available population data. We have now included a more thorough description of the parameter fitting procedure in the model parameters section: "Prior values for the biological parameters were chosen based on estimates from the literature (Moore et al 2001, Behrenfeld and Boss 2014). Parameter values are constrained to remain within realistic bounds during fitting. We tested the sensitivity of our estimates to the priors by systematically varying the initial parameter choice within the range of values reported in empirical studies. While the fitting algorithm found multiple local minima, all the biologically sensible ones cluster around the values given in Table 1." We have also specified that 84 points are used in the parameter fitting.**

**It is worth noting that the mathematical analysis in section 2 is independent of the specific parameter values and therefore shows that certain grazing functions cannot result in wintertime biomass accumulation, at least under the other model assumptions, regardless of the parameter values. However, the parameters have the same significance as in Table 1. In figure 2, we use the parameters in Table 1.**

At line 139 the authors state that mortality time scales are 10 times the scale of division rates. This is an unclear and unsupported statement (division rate can be made equivalent to a carbon fixation, or production, rate using some algebra, but I think the authors should compare like with like). The paper cited above reports it to be about 10

**REPLY: We have added additional support about the importance of grazing: "However the assumption is likely inappropriate for most blooms where grazing is a main source of mortality immediately prior to bloom formation (Calbet and Landry 2004, Irigoien et al 2005)."**

5. The dilution hypothesis is one of the possible explanations for winter accumulation of primary producer biomass, however this is based on the assumptions that light limitation is preclusive of any photon assimilation. Even if I did not expect the authors to be aware of the very recent papers on phytoplankton phenology cited above, I would have expected this work to give a more proper consideration to the role of light as a driver of phytoplankton growth. Not all the assumptions are justified, and particularly how the various processes have been averaged in the bulk mixed layer model. The authors added an implicit treatment of patchiness due to the

entrainment of phytoplankton-depleted water from the deeper ocean, but ignored and did not discuss the inhomogeneity of growth due to the exponential decay of light in the mixed layer. This implies that the authors assume that the average phytoplankton growth is determined by the average light, thus assuming negligible that the Sverdrup himself suggested that growth may take place even with strong turbulence if the phytoplankton displays a positive phototaxis. This problem has been treated extensively in Paparella and Vichi (2020), both considering cases in which the biological rates are quicker than the physical scales and the opposite, which is the main founding assumption done by Sverdrup and also made by the authors. In both cases, it has

**REPLY: There are some model assumptions, particularly the bulk mixed layer formulation, which may fail at times when mixing is weak or intermittent. However, the float profiles used in this analysis do not display significant vertical variations of chlorophyll in the mixed layer. Therefore, we assume that the biological variables are sufficiently well-mixed to justify the use of the zero-dimensional bulk mixed layer model and neglect vertical structure. This is a simplification that yields relevant insights but which necessarily does not capture all relevant aspects of the system. It is worth noting that the grazing functions are mesoscale (in the biological sense) parameterizations of microscale phenomena, a topic which is treated in the last paragraph of the discussion. These mesoscale parameterizations may be most relevant at the bulk mixed layer scale.**

**In the revised introduction we have added "This theory is based on the idea that biological and physical processes are inherently coupled. The relative timescales of mixed layer turbulence and biological growth influence the rate of phytoplankton accumulation. Phytoplankton can be released from light limitation while the mixed layer is deep if turbulence is temporarily reduced (Huisman et al 1999, Taylor and Ferrari 2011,, Paparella and Vichi 2020)." to the end of the second paragraph.**

**This is relevant to address because the essence of the theoretical contribution is that the relative dependence of growth and grazing on the mixed layer depth determines the response to dilution. The studies cited by the reviewer support the relatively weak dependence of phytoplankton growth compared to grazing on the mixed layer depth. More generally, we wish to reiterate that the contribution of our manuscript is to show that blooms triggered by deepening of a mixed layer and the associated dilution of phytoplankton can develop only for specific grazing functions. This point is treated more extensively in our general comment.**

6. Finally, I'm afraid the authors gave a somewhat biased interpretation of the discussion about the Sverdrup model and instrumentally presented it to reinforce their argumentations. It is now widely accepted that the Sverdrup model is a conceptual mathematical formulation that helped to promote the quantitative study of the ocean microbiome, in an era when marine biogeochemistry was still a branch of descriptive biology. This is clearly illustrated in the cited paper by Fischer et al, and even more in the special issue of the ICES journal edited by Sathyendranath et al. (2015). Citing from the abstract, this set of papers celebrates an elegant

and powerful hypothesis that has had long-lasting influence. Sverdrup clearly intended to describe the April-May period and not the winter accumulation, and therefore it should not be used as an alternative hypothesis for the winter case. It is also important to remark that none of the current biogeochemical models make predictions based on the Sverdrup hypothesis. As such, this is more to be seen as an emergent property, rather than an explanatory mechanism. I have thus been quite surprised to see that the authors made their own revisitation of the Sverdrup model in Sec. 2.1, attributing their interpretation to the original formulation. Sverdrup introduced just a first-order loss term and then, eventually, made a few considerations about the role of grazing, showing that the data available at that time were in agreement with his postulation. It is the authors' interpretation to attribute this constant loss term to grazing, and it should be made very clear. I would particularly recommend removing it from the title of sec. 2.1 because it would be rather misleading to those readers who are less familiar with the original paper.

**REPLY: We have removed "Sverdrup" from the title of section 2.1 and replaced it with "critical depth hypothesis". We have also changed the text in this section to distinguish between the Sverdrup 1953 contribution and the critical depth hypothesis. The section does not claim that Sverdrup was wrong, but merely incomplete. In the spring when environmental conditions change rapidly the critical depth hypothesis and Sverdrup 1953's conclusions are probably accurate. However, there are some limitations, as we state "While the critical depth hypothesis has become the most widely accepted framework to interpret the onset of spring blooms--but there are growing objections (Behrenfeld 2010)--it is not very useful to make quantitative predictions. The criterion requires knowledge of the grazing rate at the end of winter before bloom onset, which is very difficult to measure."**

2 Specific comments
L143 Concentration

**REPLY: Thank you. We have corrected this typo.**

L152 (and L155-157) A decrease in MLD would likely enhance growth terms more than the grazing losses (see point 5 above). Since episodes of restratification are actually frequent in the ocean (e.g. Smith et al., 2016), this assumption may not hold even during fully mixed conditions.

**REPLY: We selected float data that does not include lateral restratification (line 190) to avoid those effects. The statements on the cited lines refer to the specific model presented in this section and are not general statements about phytoplankton growth and physiology.**

L177 Please indicate which equation set is being used here

**REPLY: We have added "equation" to the equation reference on line 180 to make it clear which equation is being used.**

L183 More details are needed to explain the modelling of the nutrient concentration and how this has been constrained according to observations. The use of a generic nutrient variable is sensible in Sec. 2, but here, because of the connection with the specific BGC-Argo measurements, it is important to clarify which nutrient is being used and how it has been assessed in terms of seasonal cycle. The floats do not measure nitrate, but other historical data can be used.

**REPLY: The nutrient concentrations have not been constrained to observations. This is an idealized model and, as explained in subsequent sections, we fit the model to phytoplankton accumulation rates only, with no assessment of fit to nutrients or zooplankton for which no measurements are available in our dataset. We are able to convey the key points about the dependence of grazing dynamics on the mixed layer depth without explicitly fitting the model to nutrients.**

L191 Biology has been demonstrated to be patchier than physics in the North Atlantic (e.g. Mahadevan and Campbell, 2002). Please consider this in your discussion.

**REPLY: We have changed "regions" in this sentence to "trajectories", which is more precise. We have also added citations to Mahadevan et al 2012 and Karimpour et al 2018 to the previous sentence. We have discussed some of the vertical velocity processes that cause patchy productivity in the North Atlantic "Finally, there is evidence that wintertime growth can be triggered by mixed layer instabilities that occasionally restratify the mixed layer during the winter and thus increase the light available for phytoplankton (Taylor and Ferrari 2011, Karimpour et al 2018). However this cannot be the unique explanation, because float observations presented in Mignot et al (2018) and reviewed here show many examples of wintertime accumulation where these mixed layer dynamics did not seem to apply."**

L192-193. There are not many figures in this manuscript. I would thus suggest the authors to add some more information on the floats that have been used, and possibly provide a table with their identification numbers. It would be valuable to see the individual timeseries, to appreciate the phenological variability prior to their standardization.

**REPLY: These time series are available and we have added a reference to where they can be found. "All individual float trajectories are plotted in the appendix of Mignot et al. (2018)."**

L199,L202, and others. I would recommend the authors to be consistent in their terminology. There is a tendency to consider the term accumulation and growth as synonyms, however, there can be accumulation without growth.

**REPLY: We have corrected this oversight.**

L205 I would need some more explanation here. According to the description, I was expecting a non-dimensional axis and not calendar days. If the model was not rescaled, what are the t1 and t2 values for the model (and actually, having two different model formulations, which one was chosen?)

**REPLY: We have clarified by writing in the previous sentence "where t1 is the calendar day of…"**

L217 I understand that this is more of a personal choice, but I think there is value in introducing the methodology at the beginning of the paper. The parameter fitting procedure is now interrupting the presentation of the results. Also, the actual description of the optimization procedure is shown later at lines 224-227. Many of my major issues are related to the possible influence of overfitting that may have biased the results, and I think the authors should make an effort to demonstrate that this is not the case. I am not much familiar with this methodology and therefore it would be useful to know if the parameters have been tested one-at-a-time and what kind of sensitivity analysis was performed.

**REPLY: We have consolidated the presentation of the parameter fitting to a single paragraph and added additional details, as outlined in response to point 4. The parameter fitting algorithm is well-suited for both linear and non-linear problems. The optimal solution is found iteratively within the whole multi-parameter space rather than one parameter at a time. We performed a sensitivity analysis by changing the initial parameter combinations within the range of reasonable parameters.**

L231-232. Please indicate if eq. (12) was used to estimate this rate. I struggle to understand why the rate is positive but concentration keeps decreasing.

**REPLY: Yes, equation 12 was used to estimate this rate, which is a biomass accumulation rate. The concentration is decreasing because the mixed layer is deepening. Although the biomass is increasing, the concentration is still decreasing because the dilution effect outpaces the accumulation effect.**

L273-276 My concerns on this statement are expressed in the major points above. The degree of nonlinearity of the Holling type II formulation depends on the value of the parameter, which is tuned differently in the two experiments, as well as the basal loss rates. The response of zooplankton would largely depend on the extant biomass during winter, which is quite different in the two simulations.

**REPLY: We have revised this sentence to read "We demonstrated that the grazing rate as a function of phytoplankton concentration must decrease faster than linearly at low phytoplankton concentrations in order to release the phytoplankton from grazing pressure."**

L294-295 Light dependence was made less influential by assumption (see point 6 above). Please rephrase.

**REPLY: We have rephrased from "light availability" to "maximum insolation" to be more clear about what is meant here. The change in mixed layer depth is the main effect on light modification of growth.**

Comment #3
This study analyzes the importance of considering non-linear functional responses of grazing at low phytoplankton concentrations when modelling plankton dynamics. In particular, the authors point out that including these types of responses is key to reproduce the accumulation of phytoplankton biomass observed in winter in the North Atlantic. The manuscript is well written and the results and conclusions are interesting. However, I have some comments and questions that I think should be addressed in order to be published.

**REPLY: We thank the reviewer for their positive assessment of our work.**

General comments:
1) If I understood correctly, in the study the phytoplankton specific growth rate decays exponentially with depth due to light absorption with an attenuation coefficient Kd. This would mean that the response of phytoplankton growth to light only depends on the surface irradiance, the Kd, and depth; i.e. depends on the light level at a particular depth. However, it seems that this dependency is modeled as a linear response. If this is the case, please consider that P-I curves have a non-linear form, expressed as a saturating response, or a curve with an optimum due to photoinhibition (see examples in Tian 2006). Although the response might be close to linear in winter due to low irradiance levels, non-linear responses might be important later in the year.

**REPLY: It is true that a saturating irradiance model might affect the full annual cycle, especially in the spring-summer. The reviewer is correct that we assumed that the response is close to linear during the winter due to low irradiance levels. Since our focus is indeed on the winter period, we have chosen to use the linear function to reduce the number of parameters in the model. Given our focus on the winter period when light levels are low, the non-linear dependence of growth on light does not affect our core message. We have added the following sentence to the model presentation: "We model growth as a linear function of light, which reduces the number of parameters required. This choice increases the sensitivity of growth to light at high irradiance relative to a saturating model, but at the low irradiance conditions typical of the wintertime, the focus of this manuscript, growth depends approximately linearly on light (Franks 2002)."**

2) I could not find in the model how the effect of temperature on growth and grazing rates was introduced. The potential consequences of this effect were not considered in the discussion either. According to Rose and Caron (2007), low temperatures might impact more negatively

microzooplankton grazing rates than phytoplankton growth rates (although see Chen et al. 2012), which can allow phytoplankton biomass accumulation in winter. Considering this, could a combination of temperature effect and linear grazing functional response allow a phytoplankton biomass accumulation in winter? Could this combination lead to similar results as those found when applying a grazing response that is non-linear at low phytoplankton concentrations?

**REPLY: The reviewer brings up an interesting point regarding the correlation between phytoplankton concentrations and temperature. We did not include temperature explicitly in the model, but we do include a section in the discussion about possible effects of other time-dependent terms. If we assume that temperature is approximately proportional to the mixed layer depth and it only affects zooplankton grazing rates but not phytoplankton growth rates then there can be a release from grazing as the mixed deepens even when using a linear model. We have chosen to use a simplified model that does not include all potentially relevant factors to make progress towards improved understanding. This comment reveals that additional insights are likely possible using the framework that we have outlined in the manuscript. We focused on grazing because there is support in the literature for this potential mechanism for wintertime biomass accumulation (Behrenfeld 2010). There is instead inconsistent support for temperature dependence of grazing rates that would trigger an accumulation of phytoplankton in winter (Rose and Caron 2007, Lopez-Urrutia 2008, Chen et al 2012). We discuss alternative mechanisms for wintertime biomass accumulation in the paragraph beginning on line 291 where we have added an additional discussion of the potential temperature effect; "Additional factors such as temperature, which is correlated with mixed layer depth, may also have an impact on wintertime growth and grazing, representing another possible non-linear effect (Rose and Caron 2007, Lopez-Urrutia 2008, Chen et al 2012)."**

3) Using dilution experiments, Liu et al. (2021) showed that "Holling III function best described the functional response of microzooplankton grazing" and highlighted the importance o this type of response at low phytoplankton concentrations. I think this paper or similar ones based on experimental observations support the results of the current study and should be mentioned in the discussion.

**REPLY: Thank you very much for pointing out this recent reference which provides additional experimental constraints on grazing functional responses. We include a citation to this study, which supports our findings, in the discussion. In the last paragraph of the discussion we have added: "In the China Seas, the microzooplankton grazing rates are best described by a Holling type III functional response (Liu et al 2021), providing evidence for the applicability of this functional response to whole populations, at least in the low and mid latitudes."**

Specific comments:
About the title: maybe replace "An investigation of" with "Investigating" or "Analyzing."

**REPLY: Based on this comment we have simplified the title to "Grazing behavior and winter phytoplankton accumulation"**

L18-24: I think at some point here, the Critical Turbulence Hypothesis (Huisman et al. 1999) could be also mentioned as it is a famous and important one.

**REPLY: This is a good suggestion. In the revised introduction we have added "This theory is founded on the idea that biological and physical processes are inherently coupled. The relative timescales of mixed layer turbulence and biological growth influence the rate of phytoplankton accumulation. Phytoplankton can be released from light limitation while the mixed layer is deep if turbulence is temporarily reduced (Huisman et al 1999, Taylor and Ferrari 2011 , Paparella and Vichi 2020)." to the end of the second paragraph.**

L25: I would rather say that the Disturbance Recovery Hypothesis focuses on both phytoplankton growth and loss rates and how they are coupled or decoupled (i.e. on how their equilibrium is disrupted).

**REPLY: We have changed this sentence from "An alternative hypothesis proposed by Behrenfeld 2010 focuses on changes in loss rates rather than growth rates." to "An alternative hypothesis proposed by Behrenfeld (2010) focuses on changes in both loss and growth rates."**

L34: What do you mean with loss at large scales? Please elaborate. Also, I think you could include a reference for this.

**REPLY: By "at large scales" here we mean that *in situ* observations of whole populations are needed for measuring zooplankton distributions and grazing rates. This is in contrast to the widespread autonomous measurements of nutrients, light, and chlorophyll concentration. This wording is confusing and we have changed it to "for a whole population".**

L36: Loss due to grazing also depends on temperature and probably on other environmental factors (see for instance Chen et al. 2012).

**REPLY: We have revised this sentence to clarify that we did not mean that loss due to grazing depends exclusively on phytoplankton and zooplankton concentration, rather it is a choice of our work to focus on that particular dependence because its significance has not been fully appreciated. The revised sentence now reads "Loss due to grazing depends on both the concentration of phytoplankton and zooplankton populations and on the many factors that mediate the interactions between phytoplankton and zooplankton such as temperature, light, and species composition (Chen et al. 2012, Moeller et al. 2019, Strom and Welschmeyer 1991)."**

L40: It sounds like it is only possible to quantify this interaction through mathematical models. What about dilution experiments for example? It would be clearer if you say that it can be modeled through a mathematical relationship.

**REPLY: As the reviewer suggests, we have changed "quantified" to "modeled"**

Fig. 2: What are the units of the axes? Also, I am a bit confused about what the contour colors represent. At the beginning of the figure caption, it says that colors represent grazing rates and in the next sentence, it seems that colors represent the rate of change in biomass. Additionally, in the case of Holling type III, for each phytoplankton concentration, rather than a decrease in the grazing rate with deeper mixed layer depths, there is first an increase and then a decrease (i.e. It seems that there is an optimal mixed layer depth for grazing rates at each phytoplankton biomass). Finally, I think the last sentences of the caption should be better written. Decreases and increases do not occur at a particular level but rather when moving along a particular axis (see for example "This occurs at low values of phytoplankton biomass and deep mixed layers" or "At high biomass there is also a decrease in grazing rate). At a particular combination of mixed layer depth and phytoplankton biomass can occur larger/est or lower/est grazing rates.

**REPLY: Both this and the other reviewers have highlighted ways in which figure 2 is unclear. Rather than just updating the caption, in the revision we completely changed this figure to better make the main point that the dependence of the grazing rate on the mixed layer depth (and phytoplankton biomass) differs for the various grazing functional responses. To answer the specific question of the reviewer, the axis units are meters for mixed layer depth and mg C per meter squared for biomass.**

**The revised figure 2 shows the ratio of growth to grazing (non-dimensional). The theoretical results in section 2 center on the ratio of growth to grazing, so this revised figure better illustrates the main points made in that section. We have also included a more extensive caption for this figure to make it clear what is being plotted and the main points of the figure.**

L 214: Reference for Kd = 0.05 m-1?

**REPLY: The reference for this is Organelli et al 2017, which presents a global map of attenuation coefficients from Argo floats. This value is also computed for the set of floats used in this manuscript as described in Mignot et al 2018. These references are given in the revised manuscript.**

L 219-202: Maybe include a reference to Fig 3 saying between which days this peak is found.

**REPLY: We have added a reference to Fig 3.**

Fig. 3: There are too many lines in the gray grid. Select just a few for the x and y axes. Why the grazing rate is not another panel? The axis labels are too small. In the units of the axis labels,

erase the space before the exponent and separate mg and C. I'd add a vertical thicker line on day 1 to make it clear that the plots do not start on day 1. The thin black line from day 315 to day 5 is very difficult to see and can be confused with the thicker black line. Maybe use another color (blue?) and maybe make it dash.

**REPLY: We have removed the minor grid axes, increased the size of the axis labels, and made the suggested corrections to the typography of the axis labels. We have made the line from day 315 to day 5 dashed.**

L 238-239: Is it discussed whether the dp inferred by type III is more realistic? This could be supported with references.

**REPLY: We have added two sentences of discussion after this result: "One process that is included in the linear mortality in both cases is phytoplankton respiration. The linear mortality estimates from parameter fitting fall within the range of phytoplankton respiration rates from in situ observations and incubation experiments (Lopez-Sandoval et al 2014, Briggs et al 2018)."**

L246: clarify which period is the end of winter by adding in parentheses which day/s of the year (or period in days of the year).

**REPLY: We have specified the days: 320-365 and continuing to 1-75.**

Fig. 4: Separate mg and C in the axis labels. Why for one of the curves there is a labeled dot for day 135 and in the other for 130? If there is not a clear justification, use the same day for better comparison.

**REPLY: We have updated the figure to use the same days for both curves for better comparison.**

L294-295: Why does light little influence on wintertime biomass accumulation? Does not an increase in light through the seasonal cycle increase phytoplankton specific growth rates and contribute to the decoupling with their grazers?

**REPLY: It is true that light has an influence on the annual cycle, but the phenomenon of wintertime biomass accumulation does not arise from the annual cycle in light. The light is decreasing or low during the winter. Moreover, the changes in light are the same with the two functional responses.**

Technical corrections
L133: modify reference as "(Behrenfeld, 2010)" and maybe introduce it as "(see for example Behrenfeld, 2010)".

**REPLY: We have made the suggested modification.**

L136: Comma after "However".

**REPLY: We have made this correction.**

L343: very "difficult" to quantify?

**REPLY: Yes. We have added this missing word.**

---

## Referee Report (RR1)

The authors have addressed my concerns and the manuscript has generally improved.

Concerning my point regarding the description of the Holling type III equation, I have now understood what the authors mean. I have also realized that my comment was possible misleading.

In order to make the statement even clearer, I would suggest to merge line 127 (where the Holling type III equation is first described mathematically) with line 188: "the Holling type III functional response (formula) can be approximated to $\frac{g_0}{p_o} \, p^2$ at low prey concentration and asymptotically approaches a constant value for high p.."

---

## Author Response (AR2)

Reviewer #1
The authors have addressed my concerns and the manuscript has generally improved. Concerning my point regarding the description of the Holling type III equation, I have now understood what the authors mean. I have also realized that my comment was possible misleading.
*Thank you for your comments on our manuscript.*

In order to make the statement even clearer, I would suggest to merge line 127 (where the Holling type III equation is first described mathematically) with line 188: "the Holling type III functional response (formula) can be approximated to $g_0 pop^2$ at low prey concentration and asymptotically approaches a constant value for high p.."
*We have edited the sentence as suggested.*

Reviewer #3
In this new version, the authors have improved the article and properly addressed most questions and suggestions. However, I still have some comments on the manuscript.
*Thank you for your comments and careful reviews.*

General comments:

1) I agree that considering a linear response of phytoplankton growth to light is a valid simplification considering that the focus of the paper is on the wintertime, when Photosynthetic Active Radiation (PAR) levels are low. However, I think that this assumption should be discussed in more detail and treated more carefully through the paper:

*We have revised our treatment of light in this version of the manuscript. We now use a saturating response function. This change has some impact on the optimal parameters, particularly the mortality parameters. We use a saturation value of $I_0 = 40$ as in Bouman et al. 2018 and an annual cycle of light as reported in Westberry et al. 2016.*
*We would also like to note that there was an error in the parameterization of the annual cycle of light that diminished the importance of light in the previous version. We have now corrected this, but the impact on the results is minimal.*

a) In L200, it is stated that the assumption has minimal impact on the wintertime period. Until when could this impact be minimal? Until when the PAR experienced by phytoplankton (taking into account MLD and surface PAR) is low enough to assume a linear response in winter-spring? I think this should be better elaborated and justified, combining some numbers with information from the literature about P-I curves (for example, P-I curves for the North Atlantic during winter-spring).

*Using the parameters discussed above, the P-I curve is fairly linear, however we have now chosen to include a non-linear functional response because a photoadaptation parameter is implicit in a linear model.*

b) Also, although the model captures the annual cycle with this assumption, I still think that a saturating response of phytoplankton growth to light might slows down growth during the spring bloom, contributing to the recoupling between phytoplankton and zooplankton populations. This could improve the match between model predictions and observations (Fig. 3). Although I understand that not including this response simplifies the model parametrization, I still think that it should be discussed in more depth.

*We have added additional discussion of the response to light to the third discussion paragraph. "The magnitude and timing of the spring bloom and interactions between zooplankton and phytoplankton populations in the springtime may be affected by factors not considered here, such as a non-linearities in phytoplankton photophysiology."*

2) The choice of some numbers and assumptions should be better justified and/or supported with references. For example, in L112, a reference should be added in relation to the choice of the MLD definition (or a paper that uses a similar criterion). Also, in L227, I miss a reference for the choice of the N pool value of 30 mg N m-3.

*We have added a reference to Kara et al. 2000 for the mixed layer depth definition. Mignot et al. 2018 verified that the criteria works well for this region (second paragraph of the section "Data sources and processing").*
*The value of 30 mg N $m^{-3}$ was chosen based on examination of BGC-Argo float profiles in the region. We have included this information in the revised manuscript.*

Specific comments:

L5: I think this sentence should be better written. For instance, a function decreases or increases depending on the direction we are moving on the x-axis. Also, it is a bit confusing the part "in phytoplankton concentration at low concentrations". Maybe you could simplify the sentence, something like "...(or more generally non-linear) at low phytoplankton concentrations…"
*We have rewritten this sentence. It now reads "However, certain mathematical formulations of grazing as a function of phytoplankton concentration that are quadratic at low concentrations (or more generally decrease faster than linearly as phytoplankton concentration decreases) can reproduce the fall to spring transition in phytoplankton, including wintertime biomass accumulation."*

L24: Maybe introduce the sentence that starts with "Phytoplankton" with an "Also, /Additionally, " to clarify that this is a complementary hypothesis and mechanism.
*We have added "also" to this sentence.*

L46: Please replace "are modeled" by "can be modeled".
*We have made this edit.*

L75: Similar problem as in L200. What is low irradiance in numbers?
*The theoretical argument does not depend on specific parameter values and it would be a distraction to quote specific values.*

L160: This is the title, but the paragraph starts with the type II functional response. If you mean that "Grazing linear at low phytoplankton concentrations", please add this last part. Grazing was also linear in the previous section (2.1). Could you please add something to the title to make clear the difference with the previous section?
*We have changed this title to "Grazing linear at low phytoplankton concentration".*

L180: Similar to the previous case. Thus, if you mean that is quadratic in winter when phytoplankton concentrations are low, please specify.
*We have changed this title to "Grazing quadratic at low phytoplankton concentration."*

L181: The term stronger is not clear for me. I'd just indicate that is non-linear. Also, please indicate here that this happens at low phytoplankton concentrations. You indicate this later, but should be mentioned from the beginning.
*We have rephrased this sentence to address both points and it now reads "The situation is different if we prescribe a phytoplankton grazing function that decreases more rapidly than linearly as p decreases."*

L188: This is actually just one scenario of the "disturbance-recovery hypothesis", the one described in Behrenfeld (2010), where this hypothesis was initially called the "dilution-recoupling hypothesis". Later, in Behrenfeld et al. (2013), the hypothesis was extended to any environmental process that disturbs the balance between phytoplankton growth and losses and got its current name. Thus, maybe you just can simplify it to "consistent with the DRH".
*We have simplified this sentence as suggested.*

L232: I'd erase "much".
*We have made this edit*

L245: "observational timeseries", to make it clear. Also, although then it is mentioned the days of the year when this peak occurs, please include also here the days of the year that more or less delimit the peak and a reference to Fig. 3.
*We have edited this sentence by adding the word "observational" and mentioning the days of the year when the peak occurs.*

L259-260: Maybe I am missing something here but, why starting the sentence with a "Despite this"? Is it not sensible to think that if there is larger winter phytoplankton concentration using the type III grazing function (compared to type II), this grazing rate is lower?
*We changed "Despite" to "However" which does not imply an inconsistency between the high phytoplankton concentration and the low grazing rate.*

L285-288: How do you define the bloom onset? When net phytoplankton biomass accumulation is positive or when phytoplankton biomass is above a particular threshold?
*We are defining bloom onset based on the net population growth rate. We have revised this sentence to read "The deficiencies of the bulk model are evident at the spring bloom onset (period of rapid net population growth), which is slightly delayed in the model relative to the observations, occurring once the mixed layer has shoaled rather than during mixed layer shoaling."*

L309: Maybe not the best reference as this paper focuses on the "dilution-recoupling hypothesis", which describes just one scenario of the "disturbance-recovery hypothesis". Please, use Behrenfeld et al. (2013), Behrenfeld and Boss (2014) or Behrenfeld and Boss (2018). This might also be applied to the Introduction section of the paper.
*We have changed this reference to Behrenfeld and Boss 2018. We have left the reference to Behrenfeld 2010 in the introduction because that section focuses on the dilution-recoupling hypothesis.*

Figure 2: Color scales for grazing/growth are different among panels. If making the same scale for all panels is not appropriate, at least please indicate that the scales are different in the caption. In the first line of the caption, replace "(e.g" by "; e.g." or ", e.g.". Also, include spaces in the units when necessary (e.g., mg C m-3). At the end of the caption, change it by "depends also on phytoplankton concentration in the case of a Holling type II".
*We have added the sentence "Note the different color scales in each panel." We have replaced the parentheses with a comma and included spaces in the units.*
*We have not edited the last sentence of the caption because the sentence is accurate as it is written.*

Figure 3: Add in the caption "dashed thin black line".
*We have made this edit*

Technical corrections:
*We appreciate the reviewer's careful editing. We have made these editorial corrections.*

L212: Add a comma after "Mignot et al. (2018)".

L230-231: Add spaces around equal signs.

L231: Add a comma after "However".

L233: Add space before "If".

L312: Add a comma after "Furthermore".

L360: Add space after "deepen".

Bibliography:

Behrenfeld, M. J. 2010. Abandoning Sverdrup's Critical Depth Hypothesis on phytoplankton blooms. Ecology 91: 977-989.

Behrenfeld, M. J., and E. S. Boss. 2014. Resurrecting the ecological underpinnings of ocean plankton blooms. Annual review of marine science 6: 167-194.

Behrenfeld, M. J., and E. S. Boss. 2018. Student's tutorial on bloom hypotheses in the context of phytoplankton annual cycles. Glob. Change Biol. 24: 55-77.

Behrenfeld, M. J., S. C. Doney, I. Lima, E. S. Boss, and D. A. Siegel. 2013. Annual cycles of ecological disturbance and recovery underlying the subarctic Atlantic spring plankton bloom. Global biogeochemical cycles 27: 526-540.